# Sensitivity of snow models to the accuracy of meteorological forcings in mountain environment

Silvia Terzago[1], Valentina Andreoli[2], Gabriele Arduini[3], Gianpaolo Balsamo[3], Lorenzo Campo[4], Claudio Cassardo[2], Edoardo Cremonese[5], Daniele Dolia[4], Simone Gabellani[4], Jost von Hardenberg[6,1], Umberto Morra di Cella[5], Elisa Palazzi[1], Gaia Piazzi[4,7], Paolo Pogliotti[5], and Antonello Provenzale[8]

[1]Institute of Atmospheric Sciences and Climate, National Research Council, Torino, Italy
[2]Department of Physics and Natrisk center, University of Torino, Italy
[3]European Centre for Medium-Range Weather Forecasts, Reading, UK
[4]CIMA Research Foundation - International Centre on Environmental Monitoring, Savona, Italy
[5]Environmental Protection Agency of Aosta Valley, Aosta, Italy
[6]Department of Environment, Land and Infrastructure Engineering, Politecnico di Torino, Italy
[7]IRSTEA, Hydrology Research Group, UR HYCAR, 92761 Antony, France
[8]Institute of geosciences and earth resources, National Research Council, Pisa, Italy

**Correspondence:** Silvia Terzago (s.terzago@isac.cnr.it)

**Abstract.** Snow models are usually evaluated at sites providing high-quality meteorological data, so that the uncertainty in the meteorological input data can be neglected when assessing model performances. However, high-quality input data are rarely available in mountain areas and, in practical applications, the meteorological forcing used to drive snow models is typically derived from spatial interpolation of the available in-situ data or from reanalyses, whose accuracy can be considerably lower.

In order to fully characterize the performances of a snow model, the model sensitivity to errors in the input data should be quantified.

In this study we test the ability of six snow models to reproduce snow water equivalent, snow density and snow depth when they are forced by meteorological input data with gradually lower accuracy. The SNOWPACK, GEOTOP, HTESSEL, UTOPIA, SMASH and S3M snow models are forced, first, with high-quality measurements performed at the experimental site
of Torgnon, located at 2160 m a.s.l. in the Italian Alps (control run). Then, the models are forced by data at gradually lower temporal and/or spatial resolution, obtained by i) sampling the original Torgnon 30-minute time series at 3, 6, and 12 hours, ii) spatially interpolating neighboring in-situ station measurements and iii) extracting information from GLDAS, ERA5 and ERA-Interim reanalyses at the gridpoint closest to the Torgnon site. Since the selected models are characterized by different degrees of complexity, from highly sophisticated multi-layer snow models to simple, empirical, single-layer snow schemes, we
also discuss the results of these experiments in relation to the model complexity.

The results show that, when forced by accurate 30-min resolution weather station data, the single-layer, intermediate-complexity snow models HTESSEL and UTOPIA provide similar skills as the more sophisticated multi-layer model SNOW-PACK, and these three models show better agreement with observations and more robust performances over different seasons compared to the lower complexity models SMASH and S3M. All models forced by 3-hourly data provide similar skills as the
20 control run while the use of 6- and 12-hourly temporal resolution forcings may lead to a reduction in model performances

if the incoming shortwave radiation is not properly represented. The SMASH model generally shows low sensitivity to the temporal degradation of the input data. Spatially interpolated data from neighboring stations and reanalyses are found to be adequate forcings, provided that temperature and precipitation variables are not affected by large biases over the considered period. However, a simple bias-adjustment technique applied to ERA-Interim temperatures allowed all models to achieve similar performances as the control run. Irrespectively of their complexity, all models show weaknesses in the representation of the snow density.

*Copyright statement.*  TEXT

## 1  Introduction

A wide range of snow models with different degrees of complexity have been developed for hydrological applications, avalanche risk forecasting and climate studies. Some of them are also integrated within modelling chains for numerical weather forecasts or climate modelling. The degree of complexity of the snow schemes depends on the specific purpose for which they have been developed (Magnusson et al., 2015). Simple temperature-index snow models are employed in applications requiring a coarse estimate of snow depth or snow water equivalent. Physical, energy-balance, but still rather simple snow models are often used in complex modelling chains, i.e. in numerical weather prediction systems and in Earth System models, to limit the computational costs. Sophisticated multi-layer snow models are typically used to reconstruct the vertical structure of the snowpack with a high level of detail and high accuracy, as needed for avalanche warning applications.

Snow models are generally evaluated at a number of sites providing high-quality forcing and verification data. Extensive literature documents the underlying physics and the performaces of single snow models (e.g. Dutra et al., 2010; Vionnet et al., 2012; Bartelt and Lehning, 2002), and several studies compare a limited number of snow models with each other (Boone and Etchevers, 2001; Kumar et al., 2013). A few large intercomparison studies benchmarked multiple snow models, including the PILPS2d, PILPS2e, Rhone-Agg, SNOWMIP and SNOWMIP2 coordinated intercomparison projects.

PILPS2d (Slater et al., 2001; Schlosser et al., 2000) and PILPS2e (Bowling et al., 2003) aimed at evaluating snow water equivalent (SWE) simulations provided by different land surface schemes (LSS) in Russian and Swedish snow-dominated catchments, respectively. PILPS2d evaluated twentyone land surface schemes forced by 18 years of observed meteorological data from a grassland catchment in Russia, to investigate the reasons for model scatter in the output snowpack variables. Weaknesses in reproducing mid-season ablation were shown to produce systematic scatter between the models. Albedo and fractional snow cover were both key variables for an accurate representation of the amount of energy absorbed by the snowpack. The ablation during the early snow season is another major source of divergence between models: in early winter a thin snow cover is highly sensitive to changes in the forcings and the resulting differences in snowpack conditions tend to persist throughout the whole snow season if temperatures remain too cold for melt.

PILPS2e showed the difficulty of reproducing spring melting. Errors in winter snow sublimation mainly impacted the runoff simulations, while the retention of meltwater within the snowpack affected the timing of the peak in runoff rather than its magnitude. For both PILPS2d and PILPS2e the differences in model complexity did not fully explain the differences in model results.

The Rhône-AGG experiment (Boone et al., 2004) employed 15 LSSs to address the impact of the model structure and of the spatial resolution of the forcing data on the simulations of the water balance. LSSs with an explicit (bulk or multi-layer) snow scheme provided better SWE simulations than LSSs with a composite snow scheme (i.e with a mixed snow-soil-vegetation layer). LSSs with composite snow schemes showed early snow ablation and early run-off peaks compared to observations, owing to missing representation of key processes such as ripening and to inadequate representation of albedo and thermal conductivity in a mixed snow-soil/vegetation layer. SWE was strongly affected by the spatial resolution of the meteorological forcing. In fact, when high-resolution meteorological forcings were aggregated from 8 km to a coarser grid of $1°$ (about 69 km) the simulated SWE was reduced by 25-60% in 13 out of 15 LSSs. A single model explicitely considering subgrid elevation effects on the forcing was found to minimize the impact of scaling on the simulated snow water equivalent.

SnowMIP (Etchevers et al., 2002, 2004) performed an intercomparison of snow models of different complexity, used for different applications, including hydrology, global circulation models, snow monitoring, snow physics and avalanche forecasting, with the aim of identifying key processes for each application. Model complexity was found to have a strong impact on the simulation of the net longwave radiation, which strongly affects snow melt dynamics. Models relying on the explicit simulation of the internal snow processes represented snow surface temperature and the longwave radiation budget more accurately. On the contrary, model complexity had smaller impact on the net shortwave radiation, whose accuracy was dependent on the simulation of albedo. Complex models taking into account snow microstructure were able to properly represent the albedo variability (as a function of grain size and type), but simple snow models with an appropriate parameterization of albedo dynamics also guaranteed reliable estimates of this variable.

SnowMIP2 (Rutter et al., 2009) built upon SnowMIP and focused on the simulation of snowpack properties in forested areas compared to open sites, across different climatic conditions. Single models showed low correlations between different years in forested sites, and low correlations also between forested and open sites, suggesting that no single best model for all years and all sites could be easily identified. Calibration allowed reduction of root mean square error (RMSE) in forested sites but the benefits from calibration at forested sites did not transfer to nearby non-forested sites.

The mentioned studies shed light on the critical snow processes that produce the largest differences between LSS simulations. However they could not clearly define an optimal set of parameterizations for a given application, such as numerical weather predictions and climate simulations, or the minimum level of model complexity needed to achive satisfactory skills in a given application (Slater et al., 2001). A step forward in this direction was obtained by employing a single model with several options to represent each of the most snow-relevant processes, and then testing the effect of parameterizations with different degrees of complexity on the skill of the model (Essery et al., 2013; Clark et al., 2011). The best results were obtained with models having a prognostic representation of snow albedo and density, with at least a simple representation of water retention and refreezing

in the snowpack. The ongoing coordinated initiative ESM-SnowMIP (Krinner et al., 2018) is expected to provide important information on the key snow processes that should be included in Global Climate Models.

A common characteristic among past model intercomparison initiatives is the interest in testing the skills of the models in experimental sites where high-quality meteorological forcings are available, to perform a controlled evaluation of the model performances. However, such context does not represent the typical conditions occurring in practical applications, where snow models are run over large climate model grid cells, and they are coupled to atmospheric models that likely provide biased driving data (Essery et al., 2013). Moreover, reliable modelling of snowpack dynamics in mountain regions is hindered by the high spatial and temporal variability of the meteorological forcings, entailing that observations and reanalysis data at a given location are scarcely representative of the conditions of the surrounding area. A recent review paper on the European mountain cryosphere (Beniston et al., 2018) states that disentangling the uncertainties related to the model structure from those related to the meteorological input data is one of the major challenges for snow modelling at the catchment scale relevant for hydrological applications. A sensitivity analysis performed on a single, physsically-based snow model showed that the uncertainty of snow simulations due to the forcing can be comparable to or even larger than the uncertainty due to the model structure (Raleigh et al., 2015). That analysis also showed that biases in the forcing data have a larger effect than random errors. Building on the results of previous studies we now expand the perspective by considering an ensemble of snow models with different degrees of complexity and we investigate their sensitivity to the quality of the meteorological forcing, with the aim of providing information on their performances when they are forced with inputs at gradually lower temporal and/or spatial resolution.

We devised a set of experiments with six snow models with different degrees of complexity in the Alpine measurement site of Torgnon, located at 2160 m a.s.l. in Aosta Valley, Italy. First, we evaluate each model forced by accurate station measurements at 30-minute temporal resolution (we refer to this as "optimal" forcing). Second, we test the response of each model when forced by data at gradually lower temporal resolution and/or lower accuracy. To this end, we employ data from spatial interpolation of neighboring station measurements and from three gridded global reanalyses, and we extract the meteorological time series at the gridpoint closest to the Torgnon station. The site of Torgnon has been selected because it provides high-quality meteorological measurements, in particular for precipitation which is usually poorly measured in high elevation sites, and a detailed characterization of snowpack properties in terms of depth, mass and surface temperature. Moreover the Torgnon site usually experiences low wind speeds, so that the snow-drift effect is very limited. The combination of these three conditions is rare in high-elevation measurement sites but essential to reduce the uncertainties on input and validation data, and to allow for a reliable estimation of the error due to model structure. Repeating this effort at multiple test sites, for example in other alpine sites at different elevations and latitudes, or at non-alpine sites (i.e. in the Arctic) would expand the results provided by the present paper. Of course, this would come at the cost of larger uncertainties in the forcings, which propagate across the modeling exercise and complicate the interpretation and the comparison of the model outputs. For this reason, we leave this more complex investigation for a separate paper. Here we employ a multi-model and multi-forcing framework to i) assess the performances of each snow model when forced with inputs at gradually lower temporal and/or spatial resolution, ii) discuss

the relation between model performances and model complexity, and iii) provide model users with information for practical applications.

This paper is structured as follows: Sect. 2 presents the snow models employed in the study while Sect. 3 describes the station of Torgnon and the datasets employed for the experiments. Section 4 describes in detail the set of 12 devised experiments and, for each experiment, the method employed to derive the forcing. Section 5 focuses on the evaluation of snow model outputs against observations and finally Sections 6 and 7 discuss the results and draw the conclusions.

## 2 Snow models

The six models considered in this study, together with a compact overview of their characteristics, are listed in Table 1 and summarized in the following.

SNOWPACK is a highly sophisticated, multi-purpose snow and land-surface model, with a detailed description of the mass and energy exchange between the snow, the atmosphere and optionally the vegetation cover and the soil. It provides a detailed description of snow properties including weak layer characterization (Stoessel et al., 2009), phase changes and water transport in snow (Hirashima et al., 2010). A particular feature is the treatment of soil and snow as a continuum with a choice of a few up to several hundred layers (Bartelt and Lehning, 2002).

GEOTOP 2.0 is a sophisticated, snow and hydrological process-based model. Its strength is an integrated approach that takes into account the interactions between hydrological, cryospheric and geomorphological processes (Endrizzi et al., 2014). The snowpack evolution is dynamically managed by the model through a snow layering scheme which splits and merges the layers depending on their mass. The model also takes into account snow metamorphism and water percolation into the snowpack.

HTESSEL is the land-surface model of the European Centre for Medium-Range Weather Forecasts (ECMWF) Integrated Forecasting System (IFS), controlling the evolution of the snow and soil fields and the exchanges of heat and moisture between the land surface and the atmosphere above (Balsamo et al., 2009). HTESSEL includes a process-based single-layer snow scheme to represent the grid cell fraction (tile) that is covered by snow (Dutra et al., 2010). In this scheme, the snowpack is characterized by a prognostic temperature, mass, density and albedo, updated at each time step. The liquid water content is diagnosed based on the other snow fields (temperature, density and mass), allowing representation of the interception of rainfall by the snowpack and internal melting/refreezing processes (Dutra et al., 2012).

UTOPIA is a land-surface process model representing the physical processes at the interface between surface, vegetation and soil layers, including a scheme which accounts for the main processes occurring in the snowpack (Cassardo, 2015). The snowpack is considered as a single homogenous layer placed on the land surface and its mass, thermal and hydrological balances are analyzed. The model takes into account the partition of soil coverage fractions (bare soil, vegetated soil, soil or vegetation covered by snow) and is able to simulate snow water equivalent, depth, density, albedo and coverage. Snow metamorphism is parameterized.

SMASH is a two-layer snow model that reproduces some of the main physical processes occurring within the snowpack, including accumulation, density dynamics, melting, sublimation, radiative balance, heat and mass exchange (Piazzi et al.,

**Table 1.** Features of snow models in terms of model complexity following Slater et al. (2001), snow albedo ($\alpha$) parameterization, explicit representation of meltwater retention and refreezing in the snowpack ($M_w$) and a main reference.

| Snow model | Complexity | $\alpha$* | $M_w$ | Reference |
|---|---|---|---|---|
| SNOWPACK | multi-layer | 111 | Yes | Bartelt and Lehning (2002) |
| GEOTOP | multi-layer | 011 | Yes | Endrizzi et al. (2014) |
| HTESSEL | single-layer | 110 | Yes | Dutra et al. (2012) |
| UTOPIA | single-layer | 110 | Yes | Cassardo (2015) |
| SMASH | up to 3 layers | 110 | No | Piazzi et al. (2018, 2019) |
| S3M | single layer | 010 | No | Boni et al. (2010) |

*The three digits 1 and 0 represent the dependence or not of the albedo parameterization respectively on surface temperature, snow age, grainsize. 000 means fixed albedo.

2019). The model can be coupled with multivariable data assimilation schemes (Piazzi et al., 2018, 2019) allowing the joint assimilation of several snow-related observations to produce SWE and runoff estimates. To facilitate the implementation of the assimilation algorithms, the complexity of the modelling scheme is limited (e.g., liquid water storage and refreezing process are neglected). In the present study no assimilation scheme has been implemented in SMASH (open-loop configuration).

5  S3M is a spatially distributed, empirical snow model requiring only a few input variables (precipitation, temperature, incoming shortwave radiation and air humidity) to compute the water mass conservation equation and to produce a first estimate of SWE (Boni et al., 2010). A second, optional, independent estimate of the SWE field, obtained by combining spatial interpolation of surface snow depth observations and MODIS snow cover, is assimilated into the snow model using a nudging scheme. The result of the data assimilation is an updated SWE map exploiting different sources of information, modeling, remote sensing and surface stations network measurements. In the use of the model for the experiments proposed in this paper the assimilation scheme is switched off and the model runs in open-loop configuration.

In the proposed experiment all the models are used in their default configurations, so no special tuning of the model parameters is made to improve the results over the Torgnon site. All models calculate snow water equivalent and snow density as primary variables, while snow depth is derived from them.

## 3  Study site and data

### 3.1  Torgnon station data

Meteorological forcing data are provided by a fully-equipped weather observation station located at Torgnon, 2160 m a.s.l. (45°50' N, 7°34'E) in the Aosta Valley, Western Italian Alps. The experimental site belongs to the ICOS (IT-Tor, www.icos-ri. eu/) and LTER (lter_eu_it_077, www.data.ltereurope.net/deims/site) networks and it is described in detail by Galvagno et al. (2013), Filippa et al. (2015) and Piazzi et al. (2019). The location is a subalpine grassland, an abandoned pasture located a

few kilometres from the village of Torgnon. The site is characterized by an intra-alpine semi-continental climate, with mean annual temperature and precipitation of 3.1°C and 880 mm respectively (Galvagno et al., 2013). During the cold season most precipitation falls as snow and, on average, from the end of October to late May, the site is snow covered with snow depths reaching 90-120 cm (Galvagno et al., 2013). Wind-induced phenomena are limited in this site, since it experiences low winds, with an average half-hourly wind speed of $1.6 \pm 1.3$ m/s over the 2012-2014 period.

The station measures all the input variables needed to force the snow models, including air temperature, total precipitation, shortwave (SWIN) and longwave (LWIN) incoming radiation, wind speed, relative humidity, surface pressure and ground temperature at 2 cm depth (the last variable is employed by the SNOWPACK model only). These variables are measured at high frequency, and then aggregated at 30-minute temporal resolution. Precipitation measurements are performed with an OTT Pluvio2 Weighing Rain Gauge, which employs a weight-based technique to measure both liquid and solid fractions (i.e the total precipitation amount). This is a consolidated technique that provides higher confidence in the reliability of precipitation data than standard rain gauges (Kochendorfer et al., 2017a). Despite the station being equipped with a reliable pluviometer and it is exposed to low wind speeds, possible issues of precipitation undercatch can be present. The uncertainty associated with precipitation measurement has been estimated and the impact of the uncertainty of the precipitation input on snow model simulations is assessed and discussed in Appendix A. As the OTT pluviometer has been operational since mid-2012, in our analysis we consider the dataset spanning the period from October $1^{st}$, 2012 to June $30^{th}$, 2017, covering five complete snow seasons.

The Torgnon station also provides snow-related variables useful for model evaluation, including snow depth measurements, obtained by an ultrasonic distance sensor, surface temperature, snow and soil temperatures at different depths, outgoing short-wave and longwave radiation, all of them available at 30-minute resolution. Snow density and snow water equivalent are measured manually in snow pits several times per snow season during dedicated field campaigns. During the analysis period 20 manual measurements of snow density and snow water equivalent are available. Additionally, since January 2016 snow water equivalent is automatically monitored by a Campbell CS725 sensor that passively measures the attenuation of naturally existing electromagnetic radiation (Potassium-40 and Thallium-208) emitted from the soil or bedrock below the sensor. The higher the water content of the snow pack, the higher the attenuation of the radiation. The measurement is performed every 6 hours and averages the SWE over an area of about 100 m$^2$. Combining automatic snow water equivalent measurements and the corresponding snow depth measurements, additional daily snow density estimates useful for model validation have been derived for the last two snow seasons.

## 3.2 Spatial interpolation of meteorological forcings from neighboring stations

The spatial interpolation of ground meteorological observations represents one of the most commonly used practices in the operational applications of hydrological models. In order to test the performances of the models in this condition, an interpolated dataset has been generated for the Torgnon monitoring site by using the MeteoIO library (Bavay and Egger, 2014). Meteorological data from six neighboring stations have been interpolated over a squared digital elevation model of 16 km$^2$ with a grid resolution of 50 m centered on the coordinates of Torgnon (Appendix B Fig. B1 and Tab. B1). The algorithm used

for the interpolation is the inverse distance weight (IDW) as first choice for all the meteorological variables. The interpolation accounts also for vertical gradients of both temperature and precipitation, assuming constant lapse rates of -6.5°C/km for air temperature and +8.5 mm/km for precipitation. Further details are provided in Appendix B.

### 3.3 Reanalysis data

In many remote mountain areas, in-situ observations to force snow models are unavailable. In this study we explore the use of reanalysis datasets extracted at the Torgnon gridoint.

GLDAS (Global Land Data Assimilation System) is a global dataset exploiting satellite and ground-based observational data combined with advanced modelling and data assimilation techniques in order to generate optimal fields of surface variables (Rodell et al., 2004). In particular, the GLDAS-2.1 archive used in this study contains 36 land surface fields from January 2000 and updated regularly at 0.25° (lon/lat) spatial and 3-hour temporal resolutions (Rui and Beaudoing, 2018).

ERA-Interim (Dee et al., 2011) is a global reanalysis including a variety of 3-hourly surface parameters describing atmospheric and land-surface conditions, and 6-hourly upper-air parameters covering the troposphere and stratosphere. ERA-Interim has spatial resolution of 0.75°, at the latitude of Torgnon corresponding to about 59 km in the zonal and 83 km in the meridional direction. This coarse grid, which is comparable to those of state-of-the-art global climate models, implies a smooth representation of the topography and coarse information on climate variables.

ERA5 (Hersbach and Dee, 2016) is the latest ECMWF global reanalysis product, providing data at higher resolution than ERA-Interim, both in space (30 km) and in time (1 hour). ERA5 uses one of the most recent versions of the Earth system model and data assimilation methods applied at ECMWF and modern parameterizations of Earth processes compared to older versions used in ERA-Interim. With respect to ERA-Interim, ERA5 also has an improved global hydrological and mass balance, reduced biases in precipitation, and refinements of the variability and trends of surface air temperature (Hersbach and Dee, 2016).

## 4 Experimental design

We devised a set of twelve experiments at the Torgnon site employing snow models in stand-alone mode, i.e. in which the meteorological forcing is prescribed. The list of experiments is summarized in Table 2. The first experiment is a control run (CTL) in which the models are forced by optimal input data provided by the Torgnon station at 30-minute temporal resolution. This run allows testing the accuracy of the models in describing the temporal evolution of the snow-related variables in optimal conditions, namely when high-quality, high-frequency point measurements are available.

Experiments RAD-ERAI and SWIN-CLS assess the sensitivity of the models to the radiation input. As most stations, the Torgnon site is equipped with an unheated radiation sensor, which is likely to provide unreliable measurements when obstructed by snow during snowfall events. Therefore, in the experiment RAD-ERAI we take into account the shading of the radiation sensor in case of snowfall by replacing radiometer measurements with ERA-Interim reanalysis data. In the third experiment, SWIN-CLS, we employ external SWIN data resulting from the clear sky radiation (Yang et al., 2001, 2006) attenuated through the cloud masks from the Meteosat Second Generation (MSG) satellite in the following way. For each of the 34 radiometers in

**Table 2.** Overview of the experiments and their characteristics in term of forcing data, temporal and spatial resolutions and gap-filling data employed where necessary. For reanalysis datasets, the elevation of the gridpoint closest to the Torgnon station is reported.

| Experiment | Forcing | Temporal resolution | Spatial Resolution | Gapfilling |
|---|---|---|---|---|
| CTL | Torgnon station (2160 m a.s.l.) | 30' | point | ERAI* |
| RAD-ERAI | CTL except SWIN and LWIN from ERAI in case of snowfall | 30' | point | ERAI |
| SWIN-CLS | CTL except SWIN from Clearsky algorithm | 30' | point | ERAI |
| TIME-3h | Torgnon station | 3h | point | ERAI |
| TIME-6h | Torgnon station | 6h | point | ERAI |
| TIME-12h | Torgnon station | 12h | point | ERAI |
| MeteoIO | Six stations close to Torgnon (see Appendix A) | 1h | point | none |
| GLDAS | GLDAS-2.1 (2297 m a.s.l.) | 3h | 25 km | none |
| ERA5 | ERA5 (2302 m a.s.l.) | 1h | 30 km | none |
| ERAI | ERA-Interim (1480 m a.s.l.) | 3h | 80 km | none |
| ERAI-LR | ERAI, lapse-rate correction of air temperature | 3h | 80 km | none |
| ERAI-BIAS | ERAI, bias adjustment of air temperature | 3h | 80 km | none |

* % of missing values for each variable: air temperature 0.24%; surface air pressure 1%; wind speed 1.65%; total precipitation 0.25%; shortwave incoming radiation 0.33%; relative humidity 0.24%; longwave incoming radiation 0.31%.

the Aosta Valley an averaged attenuation factor $F$ is computed as:

$$F = \frac{1}{N} \sum_{i=1}^{N} \frac{R^i{}_{st}}{SWIN^i} \tag{1}$$

where N is the number of cloud-covered stations determined from the MSG cloud mask, $R^i{}_{st}$ is the measured radiation at the $i^{th}$ station and $SWIN^i$ is the corresponding modeled radiation in clear-sky condition. The incident solar radiation in cloudy

conditions at the location $j$ is given by:

$$R^j = SWIN^j F \tag{2}$$

Experiments TIME-3h, TIME-6h and TIME-12h investigate the sensitivity of the models to the temporal resolution of the meteorological forcing, since the temporal resolution of many available datasets is coarser than that employed in the CTL run. We employ the Torgnon data every 3, 6 and 12 hours since October 1st, 2012 time 00:00 UTC+01:00, and linearly

interpolate them at the 30 minute time step for all variables except for total precipitation. Precipitation is accumulated over 3, 6 or 12 hour time periods and the totals are equally distributed among the corresponding 30 minutes subperiods. Incoming shortwave radiation is linearly interpolated at the 30-minute time step for all experiments, i.e. TIME-3h, TIME-6h and TIME-12h. However, when we apply linear interpolation to derive the forcing for the TIME-12h experiment we obtain poor SWIN estimates, with a large difference between the estimated and the CTL average SWIN flux ($+97$ W/m$^2$). In order to better

estimate the SWIN forcing for the TIME-12h experiment we employ a method based on the potential (clear-sky) radiation

at 30-minute temporal resolution (Knauer et al., 2018) at the site of Torgnon. For each day of the year, the 48 values of potential radiation are rescaled according to the observed SWIN at 12:00. With this method the estimated average SWIN flux is comparable to that of the CTL forcing, with a difference of $-0.87$ W/m$^2$, showing a remarkable improvement with respect to the use of the linear method (more details are provided in Appendix C). We run the TIME-12h experiment twice,
either employing the SWIN derived from the linear interpolation method (TIME-12h-LIN) or that derived from the potential radiation method (TIME-12h-SWINPOT)

Four additional experiments, namely MeteoIO, GLDAS, ERA5 and ERAI test the case in which no surface station measurement is available and one has to rely on external data. The MeteoIO experiment employs a forcing dataset obtained through the spatial interpolation of data provided by the neighboring stations (see Sect. 3.2 and Appendix B). GLDAS, ERA5 and
10 ERAI experiments use different reanalysis products described in Sect. 3.3, namely GLDAS-2.1, ERA5 and ERA-Interim. Both MeteoIO and reanalysis data had to be rearranged and interpolated to 30-minute resolution in order to be used as forcings for snow models. In the case of ERA-Interim, for example, forecasts are initialized only twice a day at 00:00 UTC and 12:00 UTC and accumulated fluxes of total precipitation, surface solar and thermal downward radiation are available as forecasts at 3-hour intervals for the following 12 hours. From these forecasts we derive the average fluxes over 3-hour intervals and we assume
the fluxes to be constant during each interval. For the other ERA-Interim parameters, namely 2-meter temperature, dew-point temperature, surface pressure, 10 metre U and V wind components, we consider the analyses at 00:00, 06:00, 12:00, 18:00 UTC and the forecasts at +3 hours. These data are linearly interpolated in time to the integration time step (30 minutes) of the snow models. Some calculations are necessary to obtain all the variables required by the models. For example, ERA-Interim does not directly provide relative humidity, which we derive using the Magnus formula from the dewpoint temperature and the
2-meter air temperature (Lawrence, 2005).

The last two experiments, ERAI-LR and ERAI-BIAS, investigate whether bias-correcting (some of) the reanalysis drivers improves the snow model performance. To this end, we bias-correct air temperature (and indirectly the ratio of solid to total precipitation that depends on temperature) while keeping all other variables unchanged. The idea is to test whether i) the adjustment of air temperature (and the rainfall/snowfall partition) only can improve model performances and to what extent,
and ii) very simple bias correction methods can be sufficient or more sophisticated ones are necessary.

In the ERAI-LR experiment we take into account the fact that ERA-Interim has a smoothed topography and the altitude of the gridpoint closest to the Torgnon station is 680 m lower than the actual elevation of the station. In the ERAI-LR experiment we adjust the temperature data assuming a fixed moist lapse rate of 6.5°C/km. This correction results in a cooling of 4.4°C with respect to the original temperature data. In the ERAI-BIAS experiment we correct ERA-Interim air temperature using the
30 difference in the climatological averages between ERA-Interim data and the Torgnon station observations, which was found to be 2.1°C. This bias is assumed to be constant in time and it is subtracted from the original ERA-Interim temperature time series.

A desirable feature of each experiment is that the differences in the model outputs are mainly due to the internal model characteristics rather than to the different parameterizations used by the models to derive the solid and liquid precipitation fractions
from the total precipitation input. To this end, for each experiment, we estimate externally the rainfall and the snowfall amounts

using a fixed threshold on wet-bulb temperature. Specifically, precipitation is considered as snowfall when the wet-bulb temperature is lower than or equal to 1°C and as rainfall otherwise. A slightly different approach was used for GEOTOP which requires precipitation totals (rather than solid and liquid precipitation separately) and then it separates rainfall and snowfall through an internal parameterization based on a fixed threshold on dew-point temperature. In this case the dew-point temperature threshold has been calibrated to obtain approximately the same seasonal accumulated snowfall as that obtained with the method based on wet-bulb temperature. This condition is satisfied with a dew-point temperature threshold of 1.2°C. Both approaches rely on the fact that the temperature interval where rain and snow cohexists is narrower for wet-bulb temperature and dew-point temperature than for air temperature. Using the wet-bulb or dew-point temperature allows reduction of the range for which the precipitation phase is uncertain (Sims and Liu, 2015; Endrizzi et al., 2014). With this procedure all the models are driven with the same rainfall and snowfall inputs and the differences in the model simulations are assumed to depend mainly on the model structure and on the estimated snow ablation through melting, evaporation and direct air-snow sublimation (Slater et al., 2001). This procedure is applied to the total precipitation forcing of each experiment, so also to the reanalyses, even though they provide separate snowfall and rainfall among their output variables.

## 5    Results

### 5.1    CTL - impacts of the snow model structure

We run the six models driven by the best forcing available for the Torgnon site, namely the station measurements at 30-minute resolution. Figure 1 shows the simulated SWE, snow density ($\rho$) and snow depth (SD) time series provided by each model compared to the observations, over the period 2012-2017.

All the models are able to reproduce the overall variability of snow characteristics, although with different accuracy. The agreement between simulations and observations is evaluated in terms of centered pattern root mean square error, standard deviation and temporal correlation, and the resulting statistics are summarized through Taylor diagrams (Taylor, 2001) in Fig. 2. Taylor diagrams display observations as an open circle on the x-axis; the centered root mean square error between the simulated and observed variable is proportional to the distance to observations; the standard deviation of the simulated variable is proportional to the radial distance from the origin; the temporal correlation between the simulated and observed variables is shown by the angular coordinate. Evaluation metrics are calculated over simulated and observed pairs when at least one of the two values exceeds a minimum threshold, namely SWE > 0.005 m, SD > 0.01 m. Snow density pairs are compared if the corresponding values of SWE are greater than 0.005 m. The upper panels of Fig. 2 refer to the period 2016-01-01 to 2017-06-30, when continuous measurements of all three variables are available. Bottom panels refer to the full period of analysis (since 2012-10-01) for which continuous observations are available for snow depth only.

Snow water equivalent simulations are in good agreement with observations over the period 2016-2017 (Fig. 2a), although with some differences between the models. The best agreement is found with the SNOWPACK, HTESSEL, UTOPIA and GEOTOP models, showing the lowest errors (below 0.04 m SWE) and the highest correlations (above 0.85) with observa-

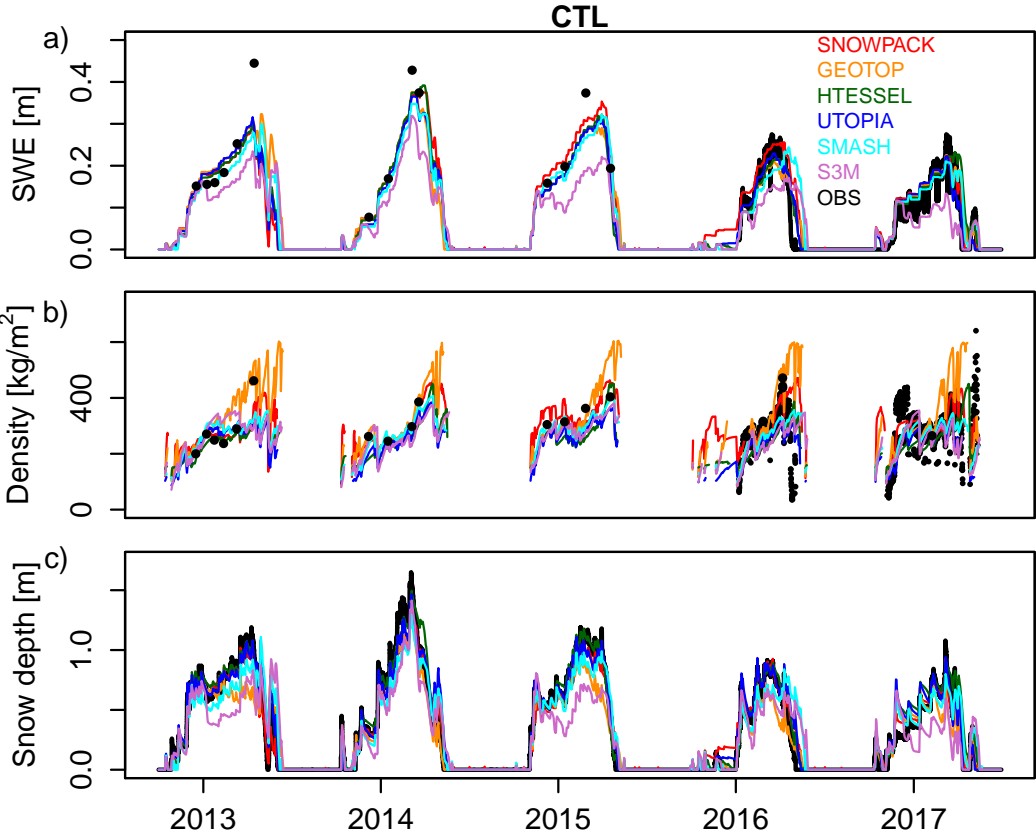

**Figure 1.** Results of the CTL experiment: a) snow water equivalent (SWE), b) snow density and c) snow depth simulated by the six models considered in the analysis, driven by optimal forcing, i.e. Torgnon station measurements at 30-minute resolution, over the period 2012-2017, compared to observations (black).

tions. SMASH and S3M are characterized by higher RMSE and lower correlation with observations with respect to the best performing models.

Snow density is simulated with lower skills compared to SWE for all models (Fig. 2b). The agreement between model simulations and observations is rather low for all models, with limited added value from highly sophisticated models. A weak
5   correlation (lower than 0.6) and large errors (above 70 kg/m$^2$) are found for both SNOWPACK and S3M, namely the most sophisticated and the simplest model, respectively. The GEOTOP model has clear deficiencies in representing spring snow density, in fact it exhibits an overestimation error increasing with time till the end of the snow season.

The ability of the models to reproduce the temporal evolution of snow depth is related to their skills in reproducing both snow mass and density. The SNOWPACK model reproduces all the three variables, namely SWE, snow density and snow
10   depth, with high scores. In the case of GEOTOP, the overestimation of spring snow density is reflected in overall lower skills in reproducing snow depth compared to the other intermediate-complexity models (Fig. 2c). In the case of HTESSEL, instead,

small errors in SWE and snow density are compensated and the model skill in reproducing snow depth is slighly higher than that of the SNOWPACK model.

The high- and intermediate-complexity models SNOWPACK, HTESSEL and UTOPIA show similar and good performances in the simulation of SWE and snow depth and they can be considered the best performing models. SMASH and S3M are characterized by higher RMSE and lower correlation with the observations, and the simplest model, S3M, shows the lowest agreement with the observations. In this experiment the model complexity is broadly reflected in the model performances, with the most sophisticated model performing best and the simplest model performing worst, likely owing to difficulties of the latter in representing snow melting (Fig. 1a). HTESSEL and UTOPIA, which are single-layer intermediate-complexity snow models performing almost as well as the most sophisticated model SNOWPACK, seem a good trade-off between model complexity and model accuracy when accurate meteorological forcing is employed.

We extend this analysis to a longer period of five complete snow seasons, from 2012 to 2017, limited to the snow depth variable. The relative skills of the models in reproducing snow depth over the full five-seasons period are very similar to those found for the last two-seasons period (Fig 2c,d). The RMSE of the models remains almost unchanged, while the correlation with observations slightly improves over the longer period. The behavior of the models is robust whether considering all the five seasons or only the last two seasons.

Figure 2e allows investigation of the variability of the model performaces in the different snow seasons compared to the whole period. SNOWPACK, HTESSEL and UTOPIA show similar skills across different snow seasons, implying robustness in reproducing a variety of conditions. Common simulation errors for several models are a positive SWE and a positive snow depth bias in the season 2015-2016 (Fig. 1a,c), when several challenging conditions occurred. First, in autumn there were isolated snowfall events separated by snowfall-free periods: mainly the SNOWPACK model, and to a lesser extent UTOPIA, failed to reproduce the rapid melting and they continued accumulating snow. Second, at the end of the snow season a very rapid melting occurred, which was not captured by any of the models. All models simulate a meltout date delayed by several days with respect to the observations. Di Mauro et al. (2018) demonstrated that the accelerated snowmelt, observed in the 2015-2016 season, was caused by the deposition of mineral dust from the Sahara: light absorbing impurities in snow, resulting from several dust deposition events, induce albedo reduction that alters the melting dynamics of the snowpack hence favouring snowmelt. As none of the models used in this study accounts for the impact of impurities on snow dynamics (and in any case no information on dust deposition is provided to the models) simulated snow melt dates in 2016 were, not surprisingly, significantly delayed.

The GEOTOP, SMASH and S3M models show different skills depending on the snow season (Fig. 2e) and they provide a wider range of variability in their agreement with the observations compared to SNOWPACK, HTESSEL and UTOPIA. For example, a season which is relatively simple to reproduce by all models is 2013-2014. An abundant but ephemeral snow cover was properly accumulated and melted by all models. After a snow-free period, the onset of a persistent snow cover was sustained by heavy snowfalls which led to the highest snow peak in the study period. After this peak, the melting has been quite steady, with few spring snowfall events. These conditions allow all models, even the simplest one, to accurately reproduce the snowpack evolution in terms of snow mass and depth. As a result, for this season the differences between the models in terms of

RMSE, standard deviation and temporal correlation with observations are smaller than for other seasons. On the contrary, the season 2012-2013 is more difficult to reproduce for some models, namely GEOTOP, SMASH and S3M, than for SNOWPACK, HTESSEL and UTOPIA. This season was characterized by many snowfall episodes of moderate and light intensity, with moderate melting in-between. In the second half of May 2013 a series of late snowfalls restored a temporary snow cover with more than 0.5 m depth that gradually melted in a couple of weeks. In these conditions, SNOWPACK, HTESSEL and UTOPIA are able to accurately represent the changes in the snow depth, while GEOTOP, SMASH and S3M generally tend to overestimate snow depth.

GEOTOP systematically overestimates snow density with increasing errors from late winter to the end of the snow season. These errors are reflected in the snow depth simulations: spring snow depth and the snow depth peak are underestimated in each snow season of the study period. SMASH, for the 2012-2013 and the 2015-2016 seasons, delays the timing of the snow depth and snow mass peaks. The delay in the representation of the snow peaks is almost fully compensated by an excessively rapid spring melting which keeps the date of ablation relatively close to the observed one. S3M systematically underestimates both snow depth and snow water equivalent during all the snow seasons, while the snow density is within the range of variability of the model ensemble. It follows that for S3M the critical variable to improve is SWE.

In conclusion, an added value of sophisticated and intermediate-complexity models compared to lower-complexity models emerges especially during snow seasons that have a more complex temporal behavior.

## 5.2  RAD-ERAI, SWIN-CLS - model sensitivity to the radiation input

A typical problem occurring in case of snowfall is that when the radiation sensors get covered with snow they record inaccurate data. To take into account this issue and test how it affects snow simulations, in the experiment RAD-ERAI we use incoming longwave and shortwave radiation data from the Torgnon station except in case of snowfall, when we employ external LWIN and SWIN data derived from ERA-Interim. In the other experiment, SWIN-CLS, we replace observed incoming shortwave radiation data with the external data described in Sect. 4. The results of these simulations are reported in Table 4. Although the difference between external data and Torgnon data can be high at the time step of the model (not shown), their overall impact on snow simulations is low. In fact, for each model we obtain values of RMSE close to those obtained in the CTL experiment. In particular, model skills do not improve using ERA-Interim or interpolated incoming radiation forcing.

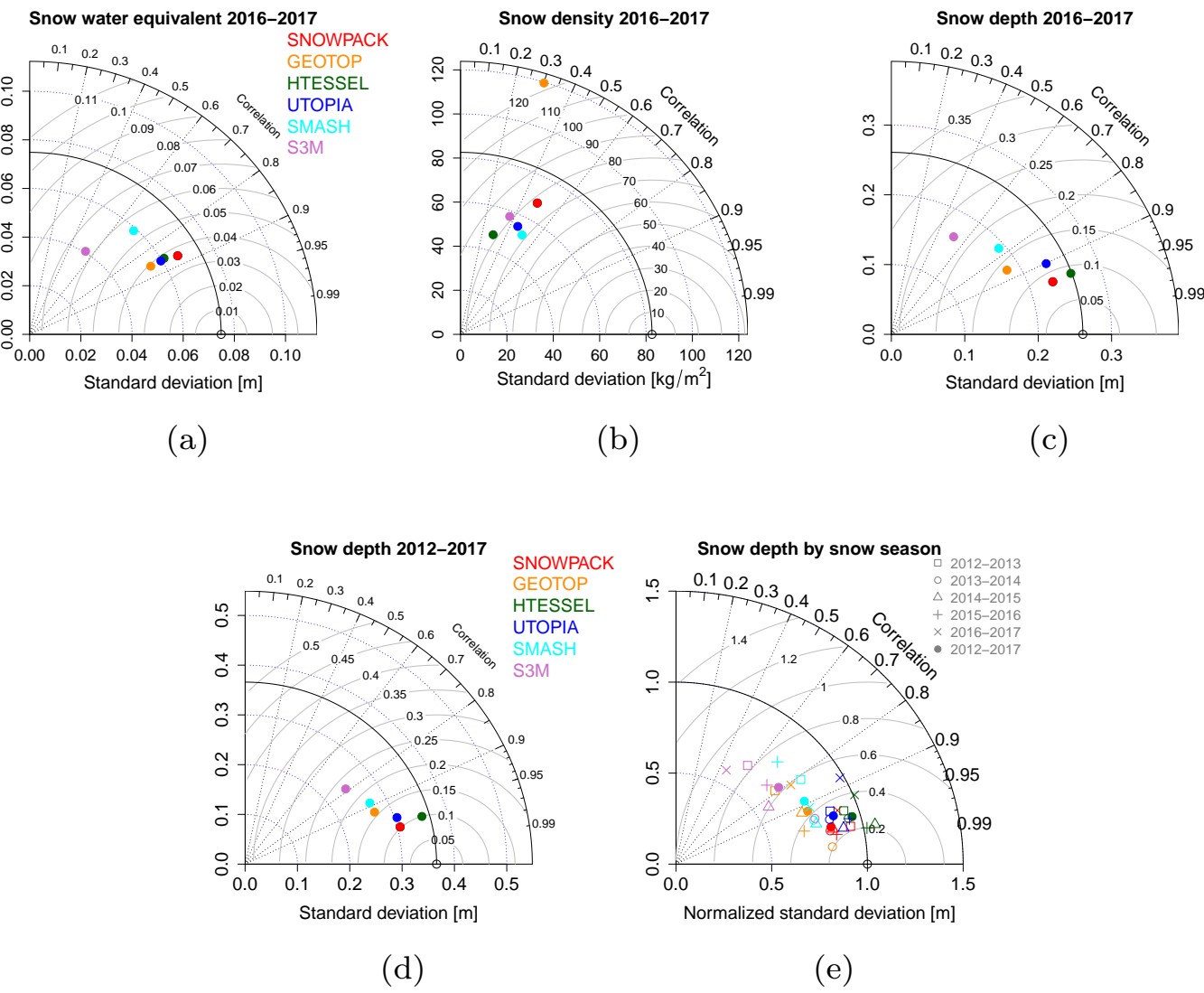

**Figure 2.** Taylor diagrams of the modeled vs. observed a) snow water equivalent, b) snow density and c) snow depth in the control experiment (CTL) for the period 2016-01-01 to 2017-06-30. Bottom panels represent the statistics of snow depth d) for the whole period 2012-2017 and e) for each snow season in the same period. Differently from other panels, in panel e) model standard deviations are normalized with respect to the observed ones.

**Table 3.** RMSE, bias and Pearson correlation of snow depth simulations with respect to observations for each model and each snow season in the control experiment (CTL).

| | | | RMSE | | | | |
|---|---|---|---|---|---|---|---|
| Model | 2012-2013 | 2013-2014 | 2014-2015 | 2015-2016 | 2016-2017 | Avg. | Std. |
| SNOWPACK | 0.08 | 0.12 | 0.10 | 0.12 | 0.08 | 0.10 | 0.02 |
| GEOTOP | 0.21 | 0.14 | 0.18 | 0.13 | 0.15 | 0.16 | 0.03 |
| HTESSEL | 0.11 | 0.13 | 0.07 | 0.08 | 0.15 | 0.11 | 0.03 |
| UTOPIA | 0.11 | 0.14 | 0.07 | 0.12 | 0.13 | 0.11 | 0.03 |
| SMASH | 0.19 | 0.19 | 0.12 | 0.23 | 0.12 | 0.17 | 0.05 |
| S3M | 0.28 | 0.20 | 0.28 | 0.21 | 0.24 | 0.24 | 0.04 |
| | | | BIAS | | | | |
| Model | 2012-2013 | 2013-2014 | 2014-2015 | 2015-2016 | 2016-2017 | Avg. | Std. |
| SNOWPACK | -0.04 | -0.03 | -0.05 | 0.10 | 0.01 | 0.00 | 0.06 |
| GEOTOP | -0.05 | -0.11 | -0.11 | -0.06 | -0.01 | -0.07 | 0.04 |
| HTESSEL | 0.04 | 0.04 | 0.01 | 0.06 | 0.11 | 0.05 | 0.04 |
| UTOPIA | 0.03 | -0.01 | -0.01 | 0.09 | 0.06 | 0.03 | 0.04 |
| SMASH | -0.04 | -0.09 | -0.04 | 0.05 | 0.04 | -0.02 | 0.06 |
| S3M | -0.08 | -0.12 | -0.21 | 0.01 | -0.10 | -0.10 | 0.08 |
| | | | Pearson Correlation | | | | |
| Model | 2012-2013 | 2013-2014 | 2014-2015 | 2015-2016 | 2016-2017 | Avg. | Std. |
| SNOWPACK | 0.98 | 0.98 | 0.97 | 0.98 | 0.94 | 0.97 | 0.01 |
| GEOTOP | 0.79 | 0.99 | 0.92 | 0.97 | 0.81 | 0.90 | 0.09 |
| HTESSEL | 0.95 | 0.96 | 0.98 | 0.98 | 0.93 | 0.96 | 0.02 |
| UTOPIA | 0.94 | 0.96 | 0.98 | 0.96 | 0.87 | 0.94 | 0.04 |
| SMASH | 0.81 | 0.95 | 0.96 | 0.69 | 0.91 | 0.86 | 0.11 |
| S3M | 0.57 | 0.95 | 0.84 | 0.74 | 0.45 | 0.71 | 0.20 |

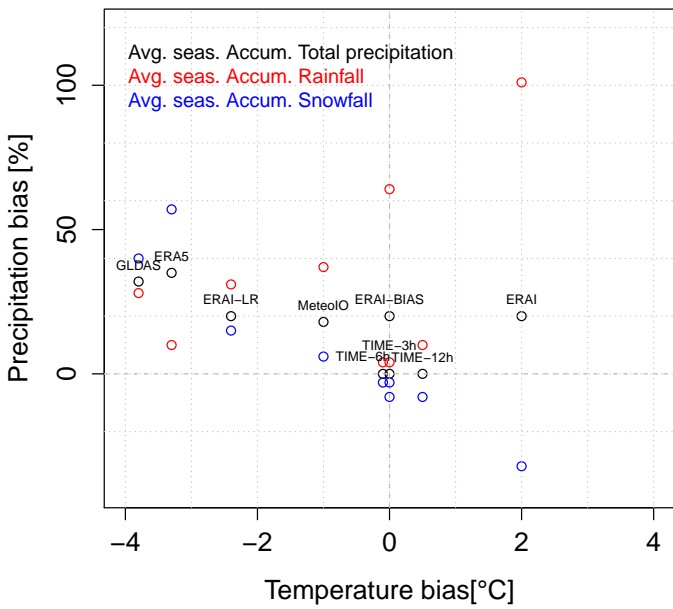

**Figure 3.** Temperature, total precipitation, rainfall and snowfall average seasonal biases in the forcings exployed in each experiment with respect to the Torgnon station measurements.

## 5.3 TIME-3h, TIME-6h, TIME-12h - model sensitivity to the temporal resolution of the forcing

A common condition when modeling snowpack evolution in data-sparse areas is the unavailability of meteorological forcings with high temporal resolution, as high as 30 minutes, like those employed in the CTL experiment. In this section we assess the sensitivity of the models to the temporal resolution of the forcing. To this aim, the original meteorological observations at Torgnon, with 30-minute resolution, have been sampled every 3, 6 and 12 hours, and then linearly interpolated at the finer (30-minute) time step, with the only exception of total precipitation that has been accumulated over the 3, 6, 12 hour periods and then equally distributed among the 30-minute sub-periods. Incoming shortwave radiation for the TIME-12h experiment has been derived with two different methods, i.e. by linear interpolating the measurements at 00:00 and 12:00 and by rescaling the potential radiation at 30 minute temporal resolution to the observed radiation at 12:00 (see Sect. 4 for details). As expected, the longer the sampling period the smoother are the input time series. For these three (and the other remaining six) experiments, we show in Fig. 3 the biases of air temperature, total precipitation, rainfall and snowfall forcings with respect to the reference forcing of the CTL experiment. Given the method employed to derive TIME-3h, TIME-6h, TIME-12h forcings we expect no bias for total precipitation, while some differences can arise in the rainfall/snowfall partition owing to possible differences in air temperature. According to Fig. 3, TIME-3h and TIME-6h air temperature biases are close to zero, while TIME-12h air temperature bias is about $0.5°C$, with the effect of reducing the amount of the solid precipitation by 10%. We investigate the impact of these biases on the snow simulations in the following.

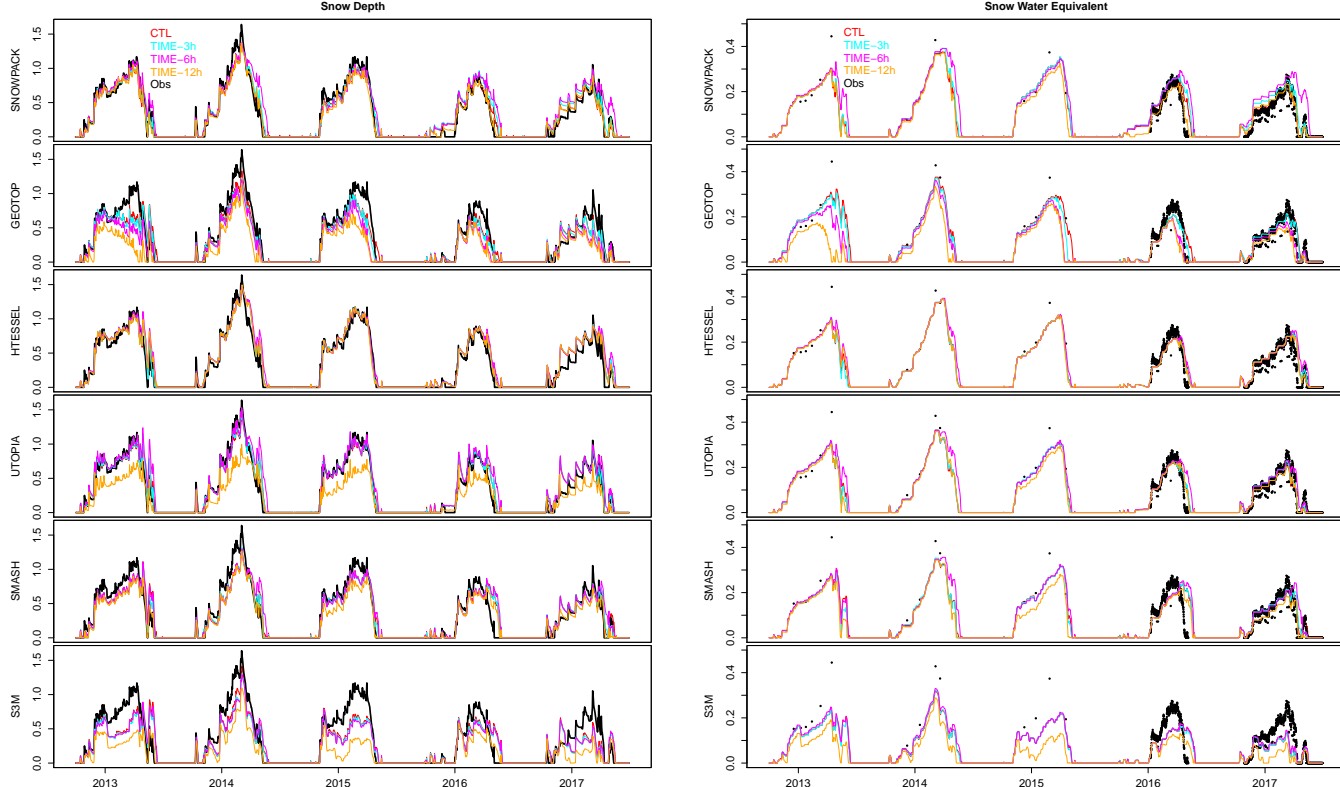

**Figure 4.** Model simulations of snow depth and SWE when the input is sampled at 3, 6 and 12 hours, compared to the CTL run and to the observations. The TIME-12h experiment employs the incoming shortwave radiation estimated with the potential radiation method.

Figure 4 represents, for all the models, the simulated snow depth and SWE when input data are sampled (or accumulated, in the case of total precipitation) at 3, 6 and 12 hours and then interpolated (or equally distributed for precipitation) over 30-minute time steps, compared to the simulated snow depth obtained with the original 30 minute resolution forcing (CTL) and compared to observations. The TIME-12h experiment employs the incoming shortwave radiation estimated with the potential radiation method; Appendix C reports the corresponding experiment employing the incoming shortwave radiation estimated with linear interpolation of the station measurements. In addition, Table 4 reports the RMSE associated with all these simulations.

The model response to the degradation of the temporal resolution of the forcing depends on the model and season. A common feature of the models is the small (or null) difference in terms of RMSE between TIME-3h and CTL simulations, indicating that using forcing data at 3-hour temporal resolution generates snow depth simulations almost as accurate as in the case of 30-minute resolution input data.

A second common feature is the general worsening of model performances when using input data at 6-hour temporal resolution, reflected in an increase of the RMSE values. TIME-6h simulations are usually very close to the CTL in the accumulation period up to the snow peak. Afterwards, in the melting period, some models, mainly SNOWPACK, UTOPIA and to a

lesser extent HTESSEL and SMASH, slightly overestimate snow mass/depth in selected seasons, contributing to an increase in the model RMSE. Compared to the TIME-6h experiment, the TIME-12h experiment with incoming shortwave radiation interpolated with the linear method (TIME-12h-LIN) shows higher RMSE on snow depth and a clear worsening of model performances (Table 4). In the TIME-12h-LIN experiment the overestimation of the incoming shortwave radiation (see Section 4) causes an underestimation of the surface snow depth. On the contrary the TIME-12h experiment with SWIN estimated with the potential radiation method (TIME-12h-SWINPOT) shows improved model performances compared to both the TIME-12h-LIN and TIME-6h experiments for SNOWPACK, HTESSEL and UTOPIA, with the former two models showing RMSE comparable with that of the CTL run. GEOTOP and S3M show similar skills in the TIME-12h experiments and higher RMSE in the TIME-12h experiments compared to the TIME-6h experiment and the CTL run. Finally the SMASH model shows little or no differences between the TIME-12h experiments and the TIME-6h, TIME-3h and CTL experiments.

In conclusion, the six models show different sensitivities to the bias in the forcing. Models with high- and intermediate-complexity (SNOWPACK, HTESSEL and UTOPIA) are sensitive to both the time degradation of the forcing and to the method used to interpolate the 12-hourly SWIN. GEOTOP and S3M are sensitive to the time degradation of the forcing but not to the method used to interpolate the 12-hourly SWIN, and finally SMASH shows low sensitivity to both the time degradation of the forcing and to the method used to interpolate the 12-hourly SWIN.

From these experiments an added value of the most sophisticated model SNOWPACK emerges. SNOWPACK forced by the 12-hour resolution forcing still provides lower errors than the simplest model S3M forced by the best available forcing at 30 minute temporal resolution (Table 4).

Concerning the TIME-12h experiment, the SWIN forcing derived with the potential radiation method provides overall better results compared that derived from linear interpolation of the station measurements. In the following the TIME-12h experiment with SWIN forcing derived with the potential radiation method will be referred to as TIME-12h.

### 5.4 MeteoIO, GLDAS, ERA5 and ERA-Interim - model sensitivity to the spatial resolution and bias in the forcing

We consider a rather standard case for which no station measurements are available for the area of interest and one has to rely on gridded datasets, which are generally characterized by lower resolution and lower accuracy with respect to station measurements. To explore a representative range of possible alternatives we employ datasets with different characteristics: the MeteoIO dataset, based on the interpolation of data from neighboring stations, the GLDAS, ERA5 and ERA-Interim reanalyses at 25, 30 and 80 km respectively. An overview of the comparison between the meteorological forcing provided by these datasets and the observations in Torgnon is shown in Fig. 3.

The MeteoIO forcing is in fairly good agreement with observations. Compared to the meteorological measurements at the Torgnon station, MeteoIO shows an average bias of $-1°C$ per snow season and about 20% overestimation of the seasonal total precipitation. However, the effect of these biases on the solid precipitation is weak, and the average seasonal snowfall is very close to the observations. When the MeteoIO forcing is used, the best agreement between simulated and observed SWE and snow depth is obtained with the GEOTOP and SMASH models. Both models provide similar RMSE values to the CTL runs. The S3M model exhibits a moderate decrease in the model performance when driven by MeteoIO compared to CTL,

**Table 4.** RMSE values associated with snow depth and snow water equivalent simulations for all models and all experiments over the periods 2012-2017 and 2016-2017, respectively.

| | RMSE snow depth [m] | | | | | |
|---|---|---|---|---|---|---|
| Exp | SNOWPACK | GEOTOP | HTESSEL | UTOPIA | SMASH | S3M |
| CTL | 0.10 | 0.17 | 0.11 | 0.12 | 0.17 | 0.25 |
| RAD-ERAI | 0.12 | 0.17 | 0.14 | 0.13 | 0.17 | 0.25 |
| SWIN-CLS | 0.11 | 0.21 | 0.12 | 0.13 | 0.18 | 0.24 |
| TIME-3h | 0.12 | 0.19 | 0.11 | 0.12 | 0.16 | 0.26 |
| TIME-6h | 0.17 | 0.26 | 0.15 | 0.18 | 0.19 | 0.27 |
| TIME-12h-LIN(SWINPOT) | 0.21(0.11) | 0.37(0.35) | 0.44(0.12) | 0.38(0.26) | 0.17(0.17) | 0.38(0.39) |
| MeteoIO | 0.23 | 0.20 | 0.38 | 0.40 | 0.19 | 0.31 |
| GLDAS | 0.67 | 0.41 | 0.79 | 0.49 | 0.63 | 0.84 |
| ERA5 | 0.74 | 0.34 | 0.76 | 0.80 | 0.71 | 0.85 |
| ERAI | 0.18 | 0.45 | 0.20 | 0.20 | 0.27 | 0.32 |
| ERAI-LR | 0.54 | 0.20 | 0.58 | 0.67 | 0.36 | 0.46 |
| ERAI-BIAS | 0.18 | 0.27 | 0.20 | 0.26 | 0.13 | 0.16 |
| | RMSE SWE [m] | | | | | |
| Exp | SNOWPACK | GEOTOP | HTESSEL | UTOPIA | SMASH | S3M |
| CTL | 0.04 | 0.04 | 0.04 | 0.04 | 0.06 | 0.08 |
| RAD-ERAI | 0.06 | 0.04 | 0.05 | 0.04 | 0.06 | 0.08 |
| SWIN-CLS | 0.05 | 0.04 | 0.04 | 0.03 | 0.06 | 0.07 |
| TIME-3h | 0.06 | 0.03 | 0.04 | 0.03 | 0.06 | 0.08 |
| TIME-6h | 0.09 | 0.05 | 0.05 | 0.05 | 0.07 | 0.07 |
| TIME-12h-LIN(SWINPOT) | 0.05(0.03) | 0.07(0.07) | 0.13(0.04) | 0.13(0.03) | 0.05(0.05) | 0.10(0.10) |
| MeteoIO | 0.10 | 0.04 | 0.13 | 0.13 | 0.07 | 0.11 |
| GLDAS | 0.38 | 0.22 | 0.38 | 0.16 | 0.33 | 0.38 |
| ERA5 | 0.28 | 0.12 | 0.22 | 0.24 | 0.26 | 0.24 |
| ERAI | 0.05 | 0.12 | 0.05 | 0.05 | 0.08 | 0.08 |
| ERAI-LR | 0.19 | 0.04 | 0.18 | 0.19 | 0.13 | 0.15 |
| ERAI-BIAS | 0.05 | 0.05 | 0.03 | 0.05 | 0.03 | 0.05 |

with lower RMSE than the HTESSEL and UTOPIA models. Conversely, the SNOWPACK, HTESSEL and UTOPIA model errors are respectively more than twice and three times the corresponding errors in the CTL run. Despite a relatively small average error in the temperature input ($-1°$C), the daily differences are generally stronger in winter and they can reach values exceeding $-4°$C. The main issue in snow model simulations is the overestimation of snow depth in winter (in selected snow seasons) and in spring (always). A plausible explanation for these errors is that colder-than-observed winter temperatures might favor the development of a cold snowpack which melts too slowly. Consequently, the models tend to overestimate the snow at the surface and to predict a delayed ablation date.

The GLDAS forcing is affected by a strong cold bias, with average temperature differences of $-3.8°$C compared to the observations, and by a moderate total precipitation bias of +32% in average over the considered seasons (Fig. 3). As expected, the large errors in the GLDAS temperature forcing lead to large errors in the simulated snow water equivalent and depth for all models, as confirmed by RMSEs in Table 4 and Fig. 6a,c,d. The magnitude of the error in snow depth shows large variations from season to season: snow depth estimates are relatively close to the observations in the first three snow seasons while a large overestimation occurs in the last two snow seasons. This behavior can be linked to the error in the total precipitation, up to +129% and +102% relative to observations in the last two snow seasons.

ERA5 has a large temperature bias ($-3.3°$C) and a moderate precipitation bias (+35%), similarly to GLDAS. The combined effect on the snowfall input is an excess of more than 50% compared to observations (Fig. 3), which clearly affects the snow model output. As expected, all models overestimate snow depth and the duration of the snow cover. The models tend to reproduce a similar evolution of snow depth as in the CTL experiment but with thicker snowpacks. In detail, SNOWPACK, HTESSEL and UTOPIA give similar snow depth outputs, consistent with the behavior found in the CTL run. GEOTOP provides the lowest RMSEs for snow water equivalent and snow depth, but this is mainly due to a compensation between the error in the ERA5 forcing (leading to overestimation) and the model error identified in the CTL experiment (leading to underestimation). In general, the difference in performance between models of different complexity is reduced when the ERA5 forcing is used. For example the RMSE is similar for the simplest model SMASH and the most sophisticated model SNOWPACK, as it is for S3M and HTESSEL or UTOPIA.

The ERA-Interim forcing (ERAI) shows a $+2°C$ temperature bias and a snowfall deficit of about 30% compared to the Torgnon observations. When forced by ERA-Interim data, GEOTOP, SMASH and S3M underestimate snow depth in all seasons, while SNOWPACK, HTESSEL and UTOPIA underestimate snow depth mainly during the season 2014-15, when the ERA-Interim snowfall is considerably lower than the observations throughout this snow season (Fig. 5a). In other snow seasons, for example 2013-14, 2015-16 and 2016-17, SNOWPACK, HTESSEL and UTOPIA snow depth simulations are in fairly good agreement with the observations (see for example Fig. 5b). Overall, SNOWPACK, HTESSEL and UTOPIA provide relatively good results when forced by ERA-Interim, with a moderate loss of accuracy with respect to the case of optimal forcing (CTL). In the following we explore the possibility to reduce the RMSE of the other intermediate- and low-complexity models by correcting the main biases in the meteorological forcings.

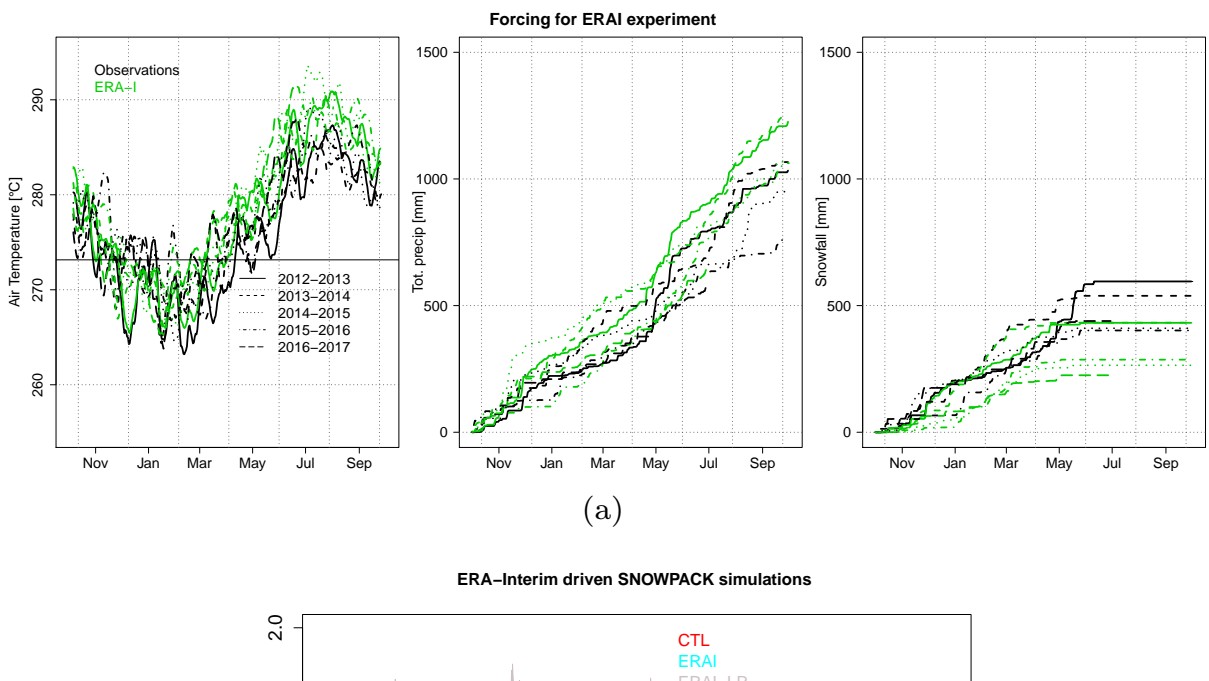

(a)

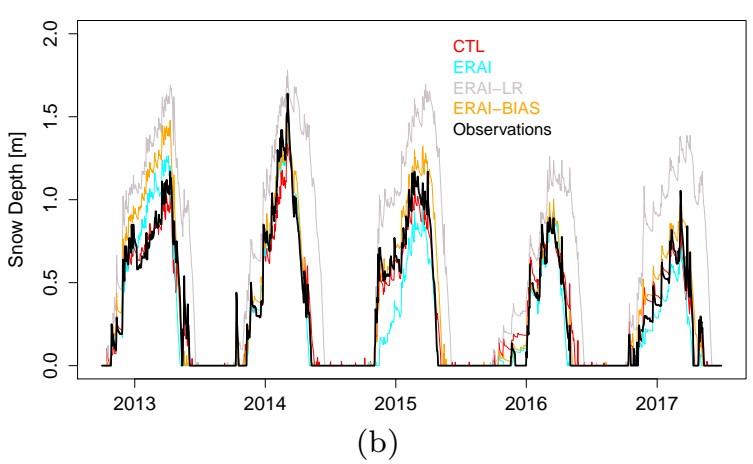

(b)

**Figure 5.** a) ERA-Interim air temperature, total precipitation and snowfall (derived as explained in Sect. 4) at the Torgnon site, compared to the station measurements (black) for each snow season of the period 2012-2017; b) Example of ERA-Interim driven snow depth simulations (ERAI, ERA-LR and ERAI-BIAS experiments), obtained using the SNOWPACK model, compared to CTL run and snow depth observations.

## 5.5 Impact of the bias adjustment of ERA-Interim air temperatures

We test the effect of two very simple bias correction techniques applied to the ERA-Interim air temperature. In the first approach, in the ERA-LR experiment, we take into account the difference in elevation between the ERAI-Interim gridpoint at Torgnon and the true elevation of this site, applying a lapse rate correction, i.e. subtracting 4.4°C from the original ERAI data. Alternatively, in the ERAI-BIAS experiment, we remove the average bias of ERA-Interim data at the Torgnon site with respect to the station measurements, i.e. subtracting 2.6°C from the original ERAI data.

The lapse rate correction excessively reduces ERA-Interim temperatures: the average temperature bias shifts from +2 to -2.4°C and the snowfall bias increases from -32 to +15% (Fig. 3).

The net effect on the model outputs (ERAI-LR experiment) is an overestimation of snow water equivalent and snow depth. With respect to the ERAI experiment, the RMSE values increase for all models except for GEOTOP, which actually shows a good agreement with observations during the seasons 2013-14 and 2015-16, while it overestimates snow depth in the first half of the other seasons. The GEOTOP underestimation error observed in the ERAI experiment is compensated by excessively cold input air temperature, which favors the development and duration of the snowpack.

The correction based on the adjustment of the mean ERAI temperature bias (ERAI-BIAS experiment) almost removes the snowfall bias. Therefore, this approach guarantees the most effective correction to improve the agreement of the forcing data with the Torgnon station measurements. Clearly this approach requires knowing at least the average temperature at the site of interest. This correction successfully reduces the RMSE in snow water equivalent and snow depth simulations with respect to the corresponding runs driven by the raw ERA-Interim data for GEOTOP, SMASH and S3M. For the most sophisticated SNOWPACK model, the correction applied to ERA-Interim data has no effects on the RMSE values of snow water equivalent and snow depth simulations, which remain unchanged. While the simulated snow depth is generally close to observations, improvements gained in the selected seasons (i.e. 2014-15) are compensated by lower performances in other (2012-13) seasons (Fig. 5d), so that, on average, the overall effect on the RMSE is negligible. For the UTOPIA model, the correction applied to ERA-Interim data has no effects on the snow water equivalent, however it slightly increases the error in snow density (lower correlation with available observations) and thus the error in snow depth simulations.

## 5.6 Discussion

While much work has been done to characterize the performances of snow models when driven by accurate input data (e.g. Vionnet et al., 2012; Boone and Etchevers, 2001; Bartelt and Lehning, 2002; Dutra et al., 2010), model responses depending on different degrees of accuracy of the input data still need to be explored in detail. This study sheds light on this research topic by assessing the simulations of six state-of-art snow models driven by input data with varying accuracy, focusing on the fully-instrumented Torgnon site, in the NW Italian Alps. The snow models selected for the analysis are characterized by different degrees of complexity, from highly sophisticated multi-layer snow models to rather simple single-layer models, with the aim of exploring relations and trade-offs between model complexity and model performances in reproducing snowpack dynamics.

In our experiment, in the case of optimal forcing, namely Torgnon station data at 30-minute resolution, the most sophisticated model SNOWPACK and the intermediate-complexity models HTESSEL and UTOPIA show the best agreement with observations. In particular HTESSEL and UTOPIA, with their single-layer, simpler snow schemes compared to SNOWPACK, can be considered a good trade-off between model complexity and model accuracy. When considering snow depth simulations, for which validation data are available for a longer period than for SWE, an added value of these high- and intermediate-complexity models compared to lower complexity models is evident, especially in the snow seasons that are more difficult to reproduce. SNOWPACK, HTESSEL and UTOPIA show similar and good performances across different seasons, revealing robustness in reproducing a variety of conditions, while the simpler snow models SMASH and S3M show larger dispersion of the seasonal scores.

Snow density is more difficult to simulate than SWE and snow depth for all models. The correlation between model simulations and observations is quite low for all models, with no clear added value from highly sophisticated ones (Fig. 2b). GEOTOP provides a much larger error compared to the other models, especially in the spring season, suggesting further checks on the snow density parameterization.

The response of the snow models forced by gradually lower accuracy data is summarized in Fig. 6, showing the model RMSE for all experiments and all variables (upper panels) and the complementary information on the model ranking (bottom panels). No remarkable differences can be detected in the model skills when using alternative radiation data instead of the Torgnon station measurements, as done in experiments RAD-ERAI and SWIN-CLS. The substantially equivalent results obtained by replacing measured data with ERA-Interim data in case of snowfall (RAD-ERAI experiment) can be explained by the combination of two conditions: the intermediate elevation of 2160 m a.s.l. and the orientation of the Torgnon site, both likely contributing to a rapid melting of the snow obstructing the radiometer. This adjustment does not affect model performances. Similar results are found employing SWIN radiation estimated as clear-sky radiation attenuated by a factor based on MSG cloud mask and neigbouring station radiation measurements (SWIN-CLS, Sect. 4). Each model shows similar RMSE in snow depth in the CTL, RAD-ERAI and SWIN-CLS experiments.

The use of accurate meteorological inputs but at lower temporal resolution, for instance Torgnon station data sampled at 3 hourly time step and then interpolated to the model time step, does not affect model performances. Similar results were obtained in a previous study in which the original forcing was averaged in time over 3 hours and the resulting time series was interpolated to the model time step (Ménard et al., 2015). Therefore we can conclude that, the typical 3-hour temporal resolution of the climate and weather forecast model outputs, as well as reanalysis data, can be suitable for driving snowpack models. The use of input data with temporal resolution lower than 3 hours requires more in-depth consideration as we observe a gradual decay of the snow model skills for most models. With 12-hourly resolution input, for example, the incoming shortwave radiation is found to be a key variable affecting the model performances. While the simple linear interpolation of the 12-hourly radiation data to the model time step provides poor SWIN estimates and poor snow model performances, a slightly more sophisticated method based on the scaling of the potential radiation on the SWIN measurements at 12:00 allows improving the snow simulations and model skills comparable to or even better than the TIME-6h experiment. With this second method the bias in the incoming shortwave radiation flux is almost completely cancelled out. A residual negative bias ($-7$ W/m$^2$) of the

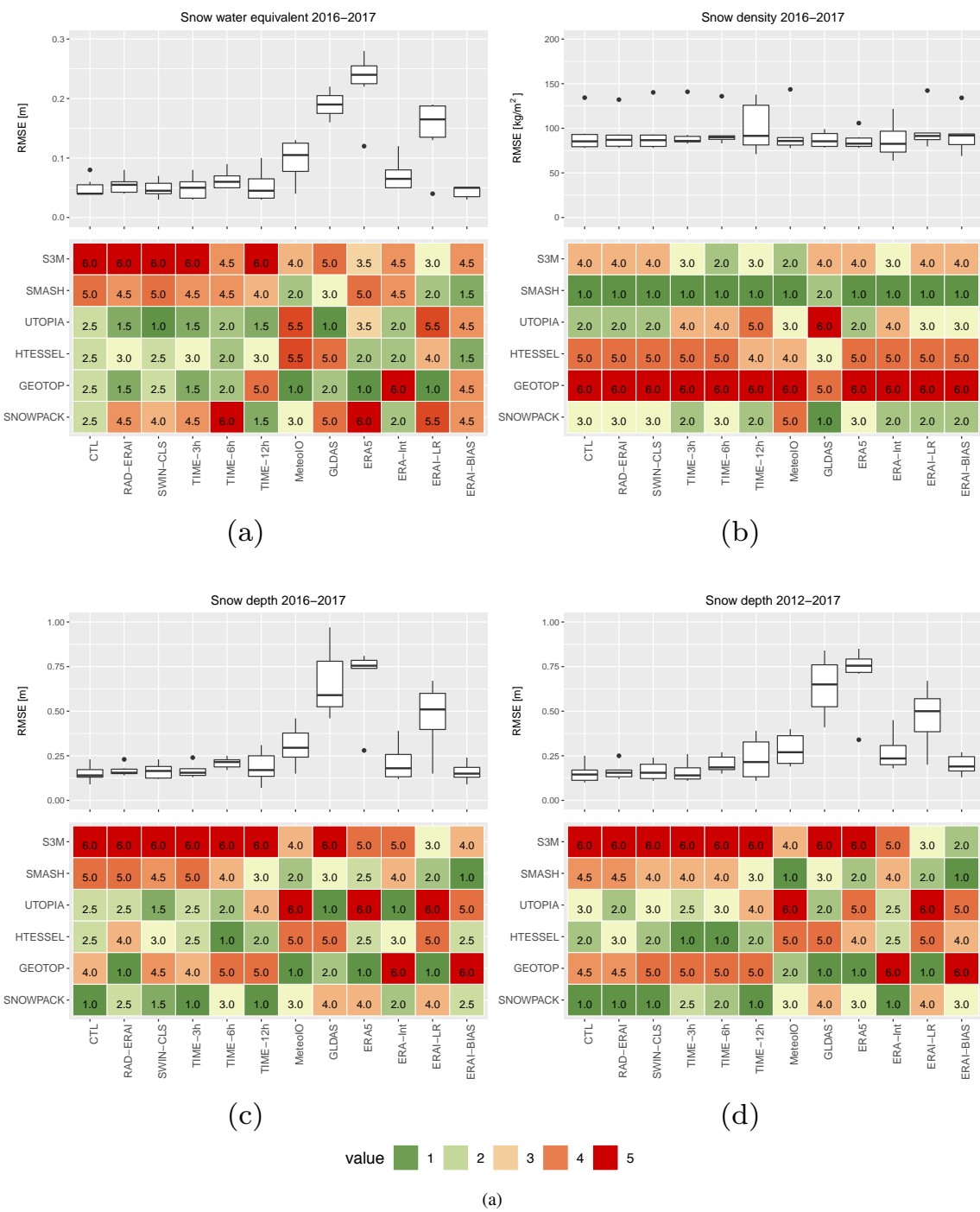

**Figure 6.** Root Mean Square Error associated to a) snow water equivalent, b) snow density and c) snow depth simulations for each experiment and each model over the period 2016-2017. Panel d) shows the same statistics as c) but on the whole period 2012-2017. Upper panels represent the boxplot statistics, lower panels represent the model rank (1=best model, 5=worst model).

incoming shortwave radiation in the TIME-6h experiment contributes to the overestimation of the snow depth at the end of the snow season. For SNOWPACK and HTESSEL the 12-hourly forcing with improved SWIN input allows surprisingly good performances, as shown by the comparable RMSEs in the TIME-12h experiment and in the CTL run.

Where meteorological station data are not available, spatial interpolation of neighboring stations data or reanalyses can be a valid alternative. In our experiment the best results are obtained with ERA-Interim forcing. Despite the coarse spatial resolution, ERA-Interim satisfactorily reproduces the meteorological conditions at the Torgnon gridpoint (Fig. 3) and the model errors in terms of snow depth and snow water equivalent are only slighly higher than in the CTL experiment (Fig. 6a,c,d). SNOWPACK, HTESSEL and UTOPIA again provide the lowest errors compared to intermediate- and low-complexity snow models (GEOTOP, SMASH, S3M). However, the latter can also be an interesting option after applying a simple adjustment of the average ERA-Interim temperature bias with respect to the Torgnon station data, and consequently also adjusting the snowfall amount. In this way the performances of the intermediate- and lower complexity snow models (GEOTOP, SMASH, S3M) can be substantially improved. The temperature adjustment based on the lapse rate (ERAI-LR), accounting for the difference in elevation between the ERA-Interim gridpoint and the real elevation of the Torgnon station, is found to worsen the model performance. In fact, this correction is blind to the local climatic features and might not be suitable in all situations. For example, in this case the lapse-rate correction is too large and it causes a temperature bias of similar amplitude but opposite sign with respect to the original ERA-Interim data. As a general remark, it is preferable to apply a temperature correction based on local temperature observations or even just climatology, when available, as the correction based on the lapse rate does not ensure a better agreement with the reference data.

Spatial interpolations of neigboring station data, such as the MeteoIO interpolation used here, can be another valid alternative in the absence of in-situ observations. In our experiment the models' RMSE values for snow water equivalent and snow depth are generally higher than those obtained using the Torgnon data at lower temporal resolution. GLDAS and ERA5 are affected, on average, by a large temperature bias and a moderate precipitation bias at the Torgnon gridpoint, probably owing to difficulties of these datasets in simulating processes in high-elevation regions. ERA5 provides slightly better performances than GLDAS. The latter has a precipitation bias that varies strongly from season to season, with large overestimation errors in the last two snow seasons (-22%,-25%,-25%,+129%,+102% respectively). By contrast, the ERA5 precipitation bias has smaller fluctuations from season to season (+34%,+16%,+38%,+47%,+41%) resulting in better and more stable performances compared to GLDAS.

The present analysis allows straighforward evaluation of the performances of each model with data of gradually lower accuracy. While, as expected also from previous studies (e.g. Jin et al., 1999; Boone and Etchevers, 2001; Luo et al., 2003; Feng et al., 2008) with accurate forcing the most sophisticated model provides the best agreement with SWE and snow depth observations and the simplest models provide the worst (Fig 6d), more heterogeneous model responses are obtained when lower accuracy data are employed. The most sophisticated model SNOWPACK is not the best performing model throughout all experiments, even though it usually ranks among the best performing ones especially in reproducing snow depth. The simplest snow model considered in the analysis, S3M, is not always the worst model, especially when low accuracy forcings are employed. SMASH shows an interesting behavior, with no brilliant performances with optimal forcing but outperforming

many other models when using lower accuracy inputs. SMASH ranks among the best performing models in the TIME-12h, MeteoIO, ERA5, ERAI-LR and ERAI-BIAS experiments, suggesting that it can be employed in data-sparse conditions with results that are comparable to those of the more sophisticated models.

The GEOTOP model provides the best snow depth estimates when forced by MeteoIO, ERA5 and ERAI-LR. However, all these forcing datasets have a cold temperature bias, and GEOTOP is affected by a systematic underestimation error in snow depth. These errors offset each other, with the effect that the RMSE in snow depth simulations is smaller than for the other models. Conversely, when using ERA-Interim forcing, GEOTOP performances are the worst owing to the positive temperature bias of the reanalysis dataset, which increases the underestimation of snow depth simulations. In this set of experiments GEOTOP shows weaknesses in reproducing the snow density and depth, thus calling for a check of its snow scheme.

The UTOPIA and HTESSEL models perform as well as the most sophisticated SNOWPACK with optimal forcing, but they require less input data, for example they do not need ground temperature. These models can be employed when no information on snowpack internal structure and stratification is needed. UTOPIA and HTESSEL provide good performances also with low temporal resolution forcings up to 6 hours and with ERA-Interim forcing. However, lower skills are found when employing the low-accuracy input dataset MeteoIO, suggesting that UTOPIA and HTESSEL can be sensitive to the bias in the meteorological forcing.

In agreement with former studies (e.g. Essery et al., 2013) also in our analysis the best performing models have i) an explicit representation of the meltwater retention and refreezing in the snowpack and ii) an intermediate-complexity representation of the snow albedo as a function of at least the surface temperature and snow age. According to our results, the representation of the snowpack as a medium with multiple layers alone does not guarantee improved results compared to models with single-layer snow schemes but able to take into account meltwater infiltration and refreezing within the snowpack.

This intercomparison exercise has been performed at a single mountain site, Torgnon, which provides ideal conditions (high-quality input and validation data, low wind speeds) to perform the sensitivity study which we aimed to. Further analysis at other sites would be useful to explore the extent to which our results could be generalized to different situations or models. We can hypothesize that the effect of the degradation in time of the forcing is probably not site-specific and similar results could be obtained in other sites (see e.g. Ménard et al., 2015). In order to assess the exportability of the results obtained in the reanalysis-driven experiments, in Appendix D we evaluate the biases of the reanalyses considered in this study (GLDAS, ERA5 and ERA-Interim) in reproducing the main drivers of the snowpack processes, i.e. temperature and total precipitation, compared to reference datasets (e.g. E-OBS version 13, Haylock et al., 2008) over the entire Greater Alpine Region (GAR, 4°E-19°E, 43°N-49°N). The time-averaged biases found at the Torgnon site are spatially consistent with those found at the mountain range scale, with the magnitude of the bias slightly varying across the region and with elevation. This analysis broadens the perspective beyond the specific case of the Torgnon site and provides guidance on the exportability of our experiment results to other areas in the Alpine region.

## 6 Conclusions

Relevant issues in snow modelling are the sparseness of meteorological stations providing all the variables required to drive and validate snow models, and the large uncertainties affecting the available measurements. Moreover, in mountain areas the spatial variability of the meteorological parameters is high, and in-situ stations could be scarcely representative of the conditions in the surrounding areas.

Currently available snow models cover a wide range of complexities, from the most sophisticated schemes that resolve the internal structure of the snowpack to the simplest ones that only provide a coarse estimate of snow depth and snow water equivalent. While several studies evaluate snow models when driven by accurate meteorological data, efforts are still needed to investigate how the models perfom when forced by lower-accuracy meteorological data, as are those typically used in mountain areas.

This study evaluates snow models of different complexities assessing their sensitivity to the accuracy of the input data. An interesting result is that some of the simplest models perform equally well or even better than sophisticated models when input data are poor. For example, the intermediate-complexity model SMASH provides lower RMSE values in snow depth simulations than many other higher-complexity models when driven by 12-hourly data, MeteoIO spatially interpolated data, GLDAS, ERA5, or the bias-adjusted ERA-Interim reanalysis. The lowest-complexity model considered in this study, S3M, provides performances that are comparable to those of the most sophisticated snow model analyzed here, SNOWPACK, when it is driven by bias-adjusted ERA-Interim data.

On the other hand, this study also shows that sophisticated snow models such as SNOWPACK can successfully reproduce snowpack variability across a wider spectrum of conditions compared to simpler snow models, outperforming them in case of isolated snowfall followed by rapid ablation. Sophisticated models provide good and more stable performances across different seasons. It is worth stressing that the most detailed snow model considered here, SNOWPACK, though not being the best performing model throughout all the experiments with lower accuracy forcings, always ranks among the best performing models at reproducing snow depth in all experiments.

Two of the intermediate-complexity snow models, HTESSEL and UTOPIA, in the case of optimal forcing provide skills in reproducing SWE and snow depth that are comparable to those of the most sophisticated model SNOWPACK. In addition, they show similar skill across different seasons, thus revealing significant robustness in reproducing a variety of conditions. HTESSEL and UTOPIA can thus be considered a good trade-off between model complexity and model accuracy in case of high-quality forcing data, while they are found to be sensitive to biases in the forcing.

Some properties which are common to all models can be highlighted: i) difficulty in reproducing snow density, especially in late spring at the end of the snow season; ii) low model sensitivity to the use of surrogate radiation input data instead of the measured ones, at least for the test site considered here; iii) comparable performances when driven by 3-hourly or 30-minute data, suggesting the possibility of using lower frequency data (up to 3 hours) without loosing accuracy in the snow output; iv) decrease of the models reliability, but not uniformly across the different models, when coarse-grid forcings are employed; v)

substantial improvement of the models performances, reducing the differences between models of different complexity, after applying a very simple bias adjustment to temperature (and consistently snowfall) forcing.

The present study has been conceived to set the basis for high-resolution modeling of mountain snow resources at the catchment and regional scales in areas where direct meteorological measurements are insufficient or unavailable and one has to rely on coarse resolution forcing. Such sensitivity experiments pave the way for the production of long-term fine-resolution reanalyses for the alpine snowpack, currently identified as a major gap for cryosphere studies (Beniston et al., 2018; Terzago et al., 2017), as well as of high-resolution future projections of the snowpack conditions. In this case snow models can be employed to refine the climate information provided by regional climate models and achieve information on snowpack characteristics at the scales required by hydrological applications, typically below 1 km. This approach, dedicated to the reconstruction of the mountain snowpack variability at fine scales is complementary to the one pursued by the ongoing ESM-SnowMIP initiative (Krinner et al., 2018) aiming at improving of the representation of snow processes and snow-related climate feedbacks in global climate models. Both approaches address issues which have been highlighted as important in cryospheric sciences (Beniston et al., 2018; Terzago et al., 2017) and provide information for a range of applications including the estimation of climate change impacts on the relevant socio-economic and environmental sectors.

*Data availability.* The datasets presented in this study can be obtained upon request to the corresponding author.

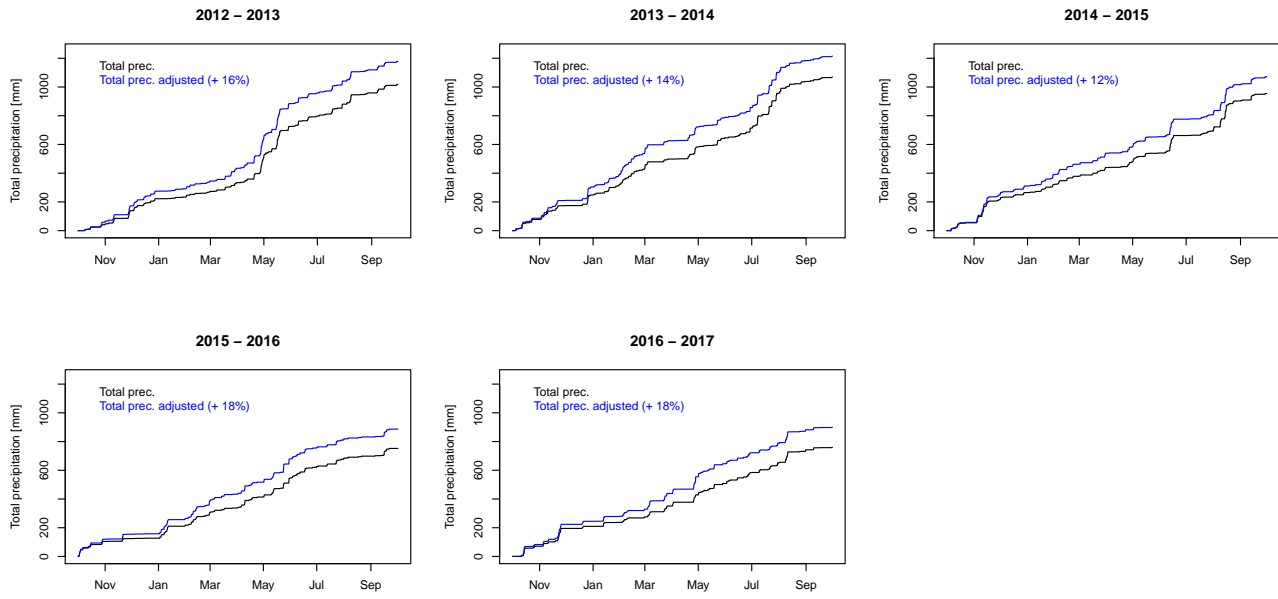

**Figure A1.** Cumulated total precipitation at the Torgnon site measured by the OTT Pluvio2 precipitation gauge (black) compared to the precipitation adjusted with the method of Kochendorfer et al. (2017a) for the five snow seasons considered (blue).

## Appendix A: Uncertainty associated with the precipitation measurements in Torgnon

We discuss here the uncertainty associated with the observed precipitation and in particular the undercatch of snow which is common in mountain areas. The primary cause for snow precipitation undercatch is related to wind speed, with the amount of precipitation measured by a precipitation gauge relative to the actual amount of precipitation decreasing with increasing wind 5  speed.

We quantified the wind-induced precipitation measurements errors by applying the method described in (Kochendorfer et al., 2017a, b). This method, derived by comparing precipitation measurements from unshielded and shielded (reference) gauges, consists in calculating a catch efficiency (CE), function of air temperature and wind speed, so that the inverse ($CE^{-1}$) can be used to correct actual precipitation data. The method has been specifically developed for OTT Pluvio2 gauges, i.e. of the same 10  type employed at the Torgnon site.

Figure A1 shows the cumulated total precipitation at the Torgnon site measured by the precipitation gauge (black) compared to the precipitation adjusted with the Kochendorfer method (blue).

The adjusted cumulated total precipitation exceeds the measured precipitation by 16% in average over the 5 snow seasons. As the correction of total precipitation directly affects the amount of solid precipitation, we tested the effects of such correction 15  on snow model simulations. We performed an additional experiment (CTL_prc-adj) in which the model forcing is the same as in the CTL run except for total precipitation, which is now the adjusted one. The snowfall fraction is then calculated from the adjusted total precipitation.

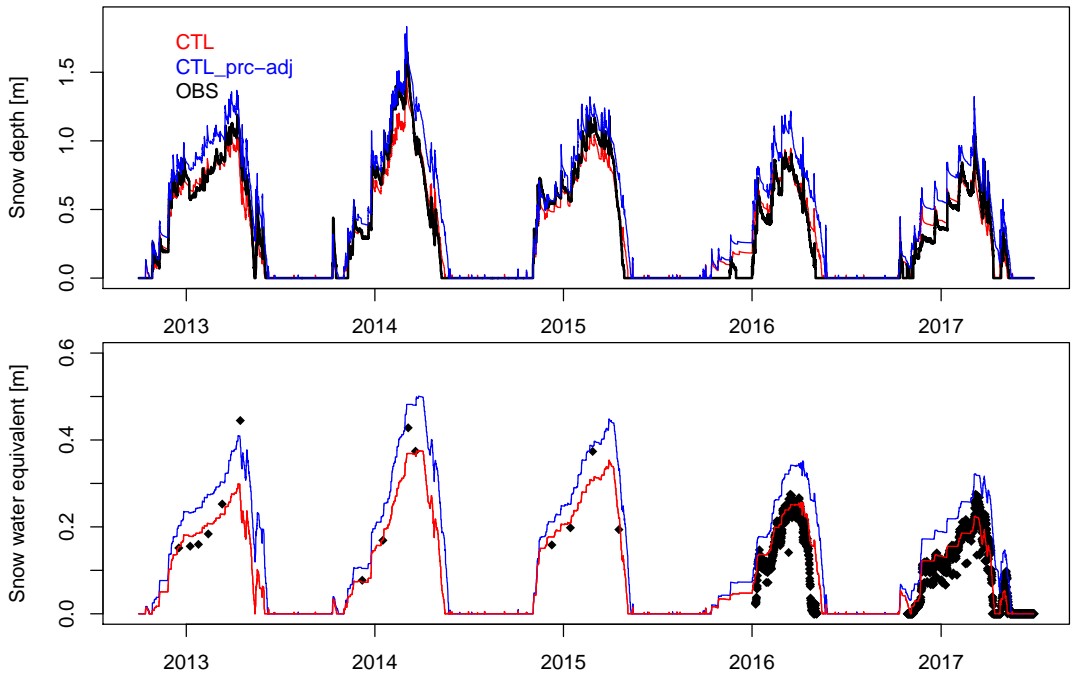

**Figure A2.** Snow depth (upper panel) and snow water equivalent (lower panel) simulated by the SNOWPACK model when the adjusted total precipitation forcing is employed (CTL_prc-adj) compared to the control run (CTL) and observations.

**Table A1.** SNOWPACK model RMSE and bias for the simulated snow depth and snow water equivalent variables in the CTL_prc-adj experiment and in the control run (CTL).

|  | Snow depth | | SWE | |
| --- | --- | --- | --- | --- |
|  | RMSE [m] | BIAS [m] | RMSE [m] | BIAS [m] |
| CTL | 0.10 | -0.001 | 0.04 | 0.02 |
| CTL_prc-adj | 0.20 | 0.170 | 0.10 | 0.09 |

Figure A2 shows the results for the SNOWPACK model, and it displays the simulated snow depth (upper panel) and snow water equivalent (bottom panel) obtained in the CTL and in the CTL_prc-adj runs compared to observations. In all snow seasons the snow depth and the snow water equivalent are remarkably overestimated in the CTL_prc-adj experiment compared with both observations and the CTL run. The additional snowfall input derived from the precipitation adjustment leads to an excess of snow accumulation on the ground which can be quantified in an average snow depth bias of 0.17 m compared to the -0.001 m bias in the CTL run. The RMSE is double in the CTL_prc-adj run compared to the CTL run (see Table A1 for details).

As the precipitation adjustment method itself is affected by its own uncertainties, and given that the application of the precipitation adjustment leads to a worsening in the snow model performances, we decided to employ the original precipitation measurements as forcing in the snow model experiments.

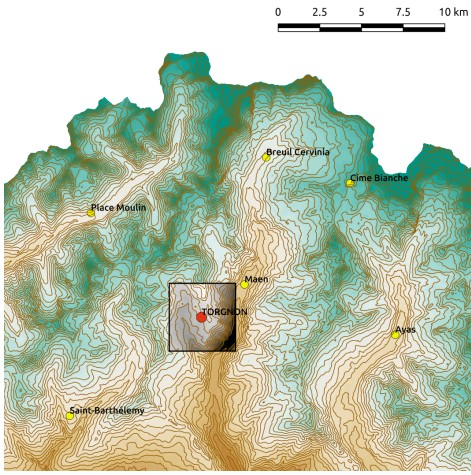

**Figure B1.** Location of the 6 neighboring stations used for producing the interpolated dataset for the MeteoIO experiment. The grey square represents the extent of the digital elevation model used for the interpolation.

**Table B1.** Characteristics of the meteorological stations used for the spatial interpolation with MeteoIO library and measured parameters: TA = air temperature; PTOT = precipitation (OTT); SWIN = incoming short wave solar radiation; VW-DW = wind speed and direction; RH = relative humidity. The stations belong to the regional meteorological network of the Aosta Valley.

| Station name | Elevation [m a.s.l.] | Distance [km] | TA | PTOT | SWIN | VW-DW | RH |
|---|---|---|---|---|---|---|---|
| Cime Bianche | 3100 | 12 | x | x | x | x | x |
| Saint-Berthélemy | 1675 | 9.8 | x | | x | x | x |
| Place Moulin | 1980 | 9.1 | x | x | x | x | x |
| Breuil Cervinia | 2000 | 10.3 | x | | | x | x |
| Maen | 1310 | 3.2 | x | | | x | x |
| Ayas | 1566 | 11.6 | x | | | x | x |

## Appendix B: Spatial interpolation of meteorological forcings from neighboring stations

In hydrological and snow modeling the spatial interpolation of ground meteorological observations is commonly employed to derive spatially continuous meteorogical forcing to drive the models. In this work, we evaluate the response of snow models with such forcing. An interpolated dataset for Torgnon monitoring site has been prepared exploiting the MeteoIO library (Bavay and Egger, 2014). The meteorological data are interpolated from six neighboring stations, over a squared digital elevation model of 16 km$^2$ with a grid resolution of 50 meters centered on the coordinate of the Torgnon monitoring site (Fig. B1 and Tab. B1).

**Table C1.** Model RMSE for the simulated snow depth in the CTL run, the TIME-12h-LIN and the TIME-12h-SWIN-POT experiments, compared to observations.

| Model | RMSE snow depth [m] | | |
|---|---|---|---|
| | CTL | TIME-12h-LIN | TIME-12h-SWIN-POT |
| SNOWPACK | 0.10 | 0.21 | 0.11 |
| GEOTOP | 0.17 | 0.37 | 0.35 |
| HTESSEL | 0.11 | 0.44 | 0.12 |
| UTOPIA | 0.12 | 0.38 | 0.26 |
| SMASH | 0.17 | 0.17 | 0.17 |
| S3M | 0.25 | 0.38 | 0.39 |

## Appendix C: The impact of the time interpolation method for SWIN in the TIME-12h experiment

We test the impact of using two different methods to derive 30 minute temporal resolution shortwave incoming radiation input when only measurements at 00:00 and 12:00 are available (as in the TIME-12h experiment). The first method is a basic linear interpolation of the available measurements. The second method is slightly more sophisticated and employs the potential (clear-sky) incoming shortwave radiation (Knauer et al., 2018) at 30 minute temporal resolution and at the coordinates of the Torgnon station, and the SWIN station measurements at 12:00. For each day of the year, the 48 daily values of potential radiation are rescaled according to the observed SWIN value at 12:00, to obtain an "estimated SWIN" (Figure C1a).

With the first method, based on the linear interpolation, the average difference between the estimated and the observed SWIN radiation over the full period is large ($+97$ W/m$^2$) while with the second method, based on the scaling of the potential radiation, the difference is close to zero ($-0.87$ W/m$^2$).

In order to test the impact of the method to interpolate SWIN radiation on snow simulations, we run two experiments in which the forcing is the Torgnon data sampled every 12 hours as explained in Section 4. The two forcings differ for the SWIN radiation input: in one case it is obtained by linearly interpolating SWIN measurements (TIME-12h-LIN) and in the other case it is obtained by rescaling the potential radiation as explained above (TIME-12h-SWINPOT).

Figure C1b shows the results of the two experiments, TIME-12h-LIN and TIME-12h-SWINPOT, compared to the CTL run and observations, for the SNOWPACK model and for the snow depth variable. The use of the SWIN forcing derived from the potential radiation leads to a remarkable improvement in the agreement with observations compared to the case when linearly interpolated SWIN is used, with the model RMSE reduced to a value which is comparable to that obtained in the CTL run (Table C1). The results for all snow models are reported in Table C1.

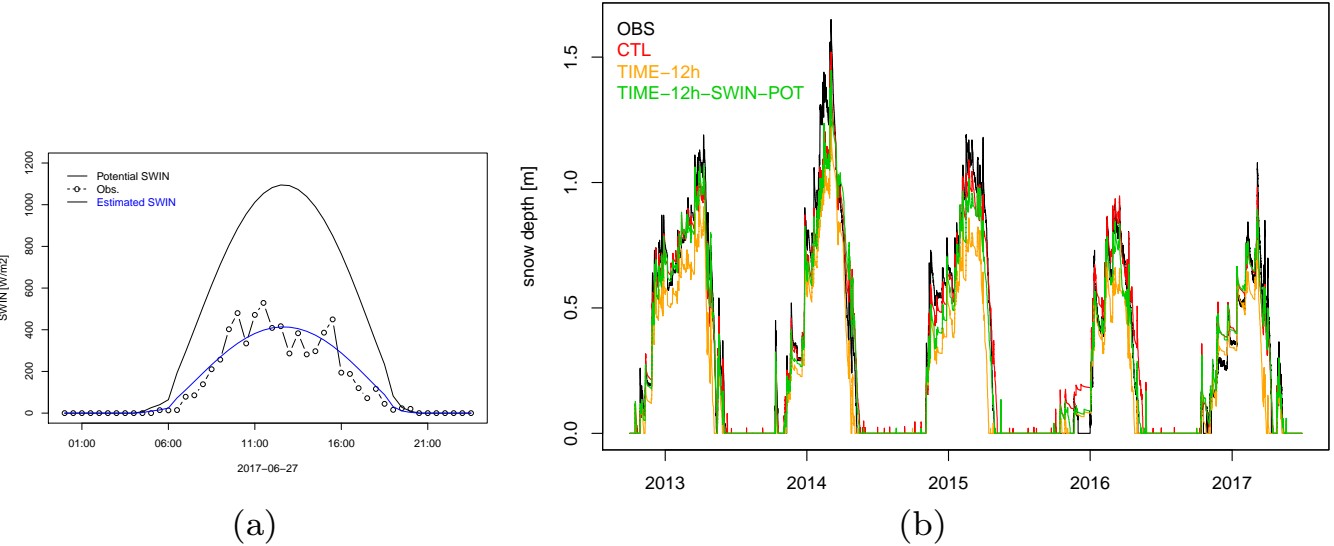

(a)              (b)

**Figure C1.** a) Measured shortwave incoming radiation (SWIN) at the Torgnon site for the day 27 June 2017 (points), potential SWIN for that day (solid black line), "estimated SWIN" from the scaling of the potential SWIN on the value registered at h 12:00; b) Snow depth simulations obtained with the SNOWPACK model for the experiment TIME-12h-SWIN-POT compared to TIME-12h-LIN, the CTL run and observations.

## Appendix D: Exportability of the results of the reanalysis-driven experiments

In order to address the issue of the exportability of the methods and results of the reanalysis-driven experiments to other areas of the Alps, we evaluated the biases of the reanalyses in reproducing the main drivers of the snowpack processes, i.e. temperature and total precipitation, compared to observational data. The aim is to evaluate the spatial distribution of the temperature and

5   precipitation biases and their consistency at the mountain range scale.

ERA5, ERA-Interim and GLDAS temperatures have been averaged over the months October-June and over the years 1980-2014 (except for GLDAS which is available since 2000 only, so the averages have been calculated over the period 2000-2014), and then compared to the observational dataset E-OBS version 13 (Haylock et al., 2008) over the Greater Alpine Region (GAR, 4°E-19°E, 43°N-49°N). E-OBS is a daily gridded data set at 0.25° resolution, based on the European Climate Assessment and

10   Data set station measurements.

ERA5 and GLDAS temperature biases are large and negative over the entire GAR (Figure D1). GLDAS bias is especially strong and it exceeds −4°C in most of the region. ERA5 bias is large at high elevations and decreases towards the lowlands. Compared to ERA5 and GLDAS, ERA-Interim temperature is in better agreement with observations, with mainly negative biases across the region and values close to zero (both positive and negative values) at the mountain ridges in Western Alps.

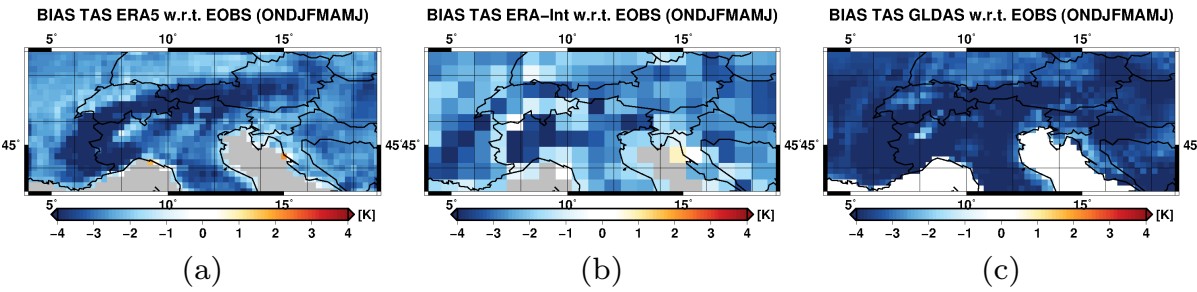

**Figure D1.** BIAS of ERA5, ERA-Interim and GLDAS air temperatures with respect to EOBS observations over the Greater Alpine Region. Temperatures have been averaged over the months from October to June and over the period 1980-2014 in the case of ERA5 and ERA-Interim, over the period 2000-2014 in the case of GLDAS.

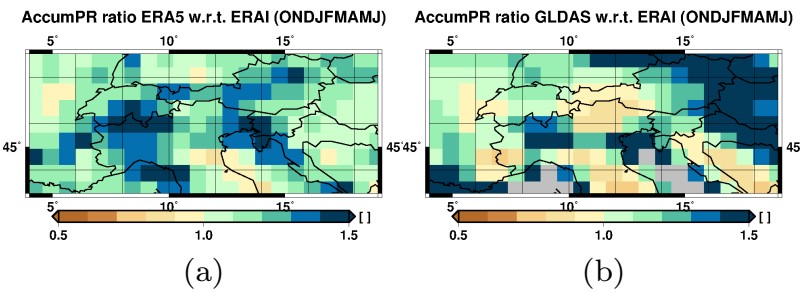

**Figure D2.** ERA5 and GLDAS relative differences with respect to ERA-Interim for the October-June accumulated precipitation over the periods 1980-2014 and 2000-2014 respectively.

All these results are consistent with those found at the Torgnon site (Fig. 3), so the biases at the point scale are reflected at the mountain range scale.

Regarding precipitation, it is well known that standard surface station gauges have problems in capturing snowfall and thus they underestimate total precipitation in mountain areas. Similarly, also observational-based datasets such as E-OBS have been found to suffer from underestimation of precipitation at high elevations (Turco et al., 2013). To overcome this problem, instead of using observation-based datasets as a reference, we evaluate precipitation ratios with respect to one of the reanalyses (ERA-Interim), since reanalyses inherently take into account orographic effects. Figure D2 shows the ERA5 and GLDAS October-to-June accumulated precipitation ratios relative to ERA-Interim over the periods 1980-2014 and 2000-2014 respectively (GLDAS is available since 2000). Also in this case ERA5 spatial pattern is homogeneous over the Alpine range, with ERA5 showing consistently more precipitation than ERA-Interim in the mountain areas. GLDAS precipitation is found to be in slightly better agreement with the ERA-Interim reanalysis than ERA5, with relative precipitation bias close to 1 over the Alpine range.

Overall, this analysis providing information on the spatial variability of the temperature and precipitation biases in the reanalysis products over the Alpine region broadens the perspective beyond the specific case of the Torgnon site. The time-

averaged biases at the Torgnon site are spatially consistent with those found at the mountain range scale, with the magnitude of the bias slightly varying across the region and with the elevation. Similar biases in the forcing suggest that the methods applied in the reanalysis-driven experiments could be extended to other sites in the Alps and could lead to results not too dissimilar from those found at Torgnon.

*Author contributions.* ST, AP, CC, EC, SG, UMC, PP conceived the idea of the experiments. All authors participated in the collection of the meteorological datasets for the experiments. ST, VA, GA, LC, DD, GP, PP performed the simulations. ST analyzed the simulations and prepared all figures, all authors provided support in the interpretation of the results. ST wrote the paper with support from all authors.

*Competing interests.* The authors declare that no competing interests are present.

*Disclaimer.* TEXT

*Acknowledgements.* This work received funding from the Italian Project of Interest NextData of the Italian Ministry for Education, University and Research and from the European Union's Horizon 2020 research and innovation program under grant agreement no. 641762 (ECOPOTENTIAL). Part of this work was performed in the framework of the MEDSCOPE (MEDiterranean Services Chain based On climate PrEdictions) ERA4CS project (grant agreement no. 690462) funded by the European Union. Jost von Hardenberg acknowledges support from the European Union's Horizon 2020 research and innovation programme under Grant agreement 641816 (CRESCENDO).

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
