# Peer review of "Sensitivity of snow models to the accuracy of meteorological forcings in mountain environment"

_Hydrology and Earth System Sciences, 2019_

## Referee Comment (RC1) · Anonymous Referee #1 · 20 Dec 2019

General comments:

The authors of the manuscript (ms.) have tested six non-calibrated snow models at one mountain location by varying the time-resolution and origin of model forcing. The quality of meteorological forcing is indeed an important element in snow modelling, and the authors have here examined the sensitivity of snow model performance to varying input data quality. The manuscript is quite well-written, and the illustrations are mostly clear and understandable. The research idea is based on straightforward model testing, i.e. running an ensemble of models with different forcing data and calculating statistics on how the models' performance (evaluated against point snow observations) vary. However, the two main weaknesses of the ms. are in my opinion that 1) the model evaluation is made at only one single site (Torgnon), and that 2) there is no discussion

or treatment of uncertainty in solid precipitation measurements, which directly affect snow amounts in the comparison. Although the ms. may provide an interesting case-study for those interested in the specific models and the local site, the ms. does not have, in my opinion, large enough impact and interest for a wider audience to warrant publication in HESS. After reading the ms. twice, I felt in the end that I did not get much relevant information out of it for my modelling work, in another place, another country. As the authors themselves state (p. 22, lines 8-9): "This study offers some hints on this research topic".

Specific comments:

* As snow is often spatially very inhomogeneously distributed, normally the utility in snow modelling is to get a grasp of this spatial variability. Thus, simulating snow in just one point has limited relevance, mostly restricted to snow process studies. In another point, the authors' results and model ranks might be changed significantly. As the authors themselves state, on p. 2, line 15: "Snow models are generally evaluated at a number of sites"; on p. 26, lines 2-3: "Further analysis at other test sites would be useful to explore the extent to which our results could be generalized to different situations or models".

* The authors note on p.4 line 1-12: "the uncertainty on snow simulations due to the forcing can be comparable to or even larger than the uncertainty". They also refer on p.7 line 2 to Kochendorfer et al. (2017), who assess and provide algorithms to deal with the undercatch of solid precipitation. However, no effort is made to discuss, assess or correct the precipitation measurements at Torgnon station for the undercatch and/or examine the sensitivity of the authors' results for the inherent uncertainties in the observation-based model precipitation input (their CTL experiment with "optimal forcing").

* The authors claim that a bias adjustment of forcing data leads to more precise results (p.9. lines 29-31). This seems to me like a rather trivial point.

\* The linear interpolation of shortwave radiation e.g. in the TIME-12h case, causing the large deviations of +97 W/m2 (p. 18, lines 3-4), is an unrealistically simplistic way to make the interpolation. In real modelling practice, I suppose most of us would use a sinus-curve form or something like that. Consequently, the issue here is more of poor modeling practice than lower time-resolution. This is only mentioned in section Discussions (p.24, line 10), but would have been best to put into practice already in the authors' study.

\* The point-specific biases and errors of the MeteoIO, GLDAS, ERA5 and ERA Interim, described in Sections 5.4 and 5.5 for the single Torgnon site, emphasize the weakness of this case study: the model evaluation results are difficult to generalize outside this site, where things and biases could be very different. Also, compensating errors may occur in the models which improve model performance. In other words, one gets "right results for wrong reasons" (p.13, lines 14-16; p. 21, lines 4-5; also p.24, line 17).

\* Normally, the authors' results would be compared to previous studies in the section "Discussion". However, this three-page section only has one (!) single reference to other published studies. It also repeats many things already pointed out previously in the ms.

Other points:

\* p.7 lines 22-23: provide values for the vertical gradients used here.

\* Fig. 2. The meaning of dashed line circles in the Taylor graph should be explained.

\* The Appendix is short (a little figure and table, and five lines of text) and could be easily added to main text in Section 3.2.

\* The ms. contains a lot of numbers in tables 3-4, and displaying them more reader-friendly, like in the nice Fig. 6, would be good.
* * *
511, 2019.

---

## Author Comment (AC1) · 20 Dec 2019

Dear Referee,

thank you for your comments. A comprehensive reply to all the points you raised will be prepared after the closure of the public discussion. Here we reply specifically to the main limitation highlighted in your report, i.e.:

" The research idea is based on straightforward model testing, i.e. running an ensemble of models with different forcing data and calculating statistics on how the models' performance (evaluated against point snow observations) vary. However, the two main weaknesses of the ms. are in my opinion that 1) the model evaluation is made at only one single site (Torgnon) [...]"

[Figure]

We see your point and we are aware that the choice of a single study-site can be highlighted as a limitation, however we have some motivations to support this choice. The strengths of our work are, in our opinion, the analysis of a multi-model ensemble, representative of different degrees of complexity of snow models, and the analysis of a wide range of possible meteorological forcing datasets, to explore in detail the response of the models to forcings with different characteristics and resolutions in time and space. When planning this large, collaborative experiment we carefully considered the choice of the site where to perform our analysis. The site we finally selected is quite unique as it provides high quality data in particular for precipitation (in most cases poorly measured in high elevation sites) and it is affected by low wind speeds, so that the snow-drift effect is limited. The combination of these two conditions is rare in high-elevation mountain measurement sites, nevertheless it is essential if we want to reduce the uncertainties on the input data. Repeating this effort in multiple test sites, for example in other Alpine sites at different elevations, or at non-alpine sites, i.e. in the Arctic) would certainly expand the vision provided by the paper but at the cost of larger uncertainties in the forcings which propagate across the modeling exercise and complicate the interpretation of the model outputs. Reducing the uncertainty on the "control" forcings is a prerequisite, in our case, to better separate the error due to model structure from the error due to the forcing. In this context, the selected site represents an appropriate benchmark for the aim of the paper. Extending the investigation to other test sites with less "optimal" forcing is of great interest but in our opinion it deserves attention in a separate paper, as a more complex framework is needed compared to the one adopted in this study.

We hope to have clarified the motivations underlying our choice. We believe that this study, shedding light on the impacts of the model complexity and of the accuracy of the forcing on snow simulations, could be of interest for the readers of Hydrology and Earth System Sciences involved in catchment hydrology, snow modelling and snow and water resources management.
Many thanks for your feedback and kind regards.

Silvia Terzago and co-authors

---

## Referee Comment (RC2) · Richard L.H. Essery (Referee) · 27 Dec 2019

Although limited to a single site and short on mechanistic explanations, the evaluations of several models in simulating snow mass, depth and density with several forcing datasets in this paper are of value (another with similar aims that should be cited is https://journals.ametsoc.org/doi/full/10.1175/JHM-D-15-0013.1). Measurements of outgoing shortwave and longwave radiation are mentioned but not used in the model evaluations; these might provide more insight.

p2, line 28

There is no need for feedbacks for early differences in snow simulations to persist throughout the winter if the imposed conditions remain too cold for melt.

[Figure]

p3, line 22

delete "air" in "open air sites"

p3, line 25

Specifically, Rutter et al. (2009) found that benefits from calibration at forest sites did not transfer to nearby non-forested sites. Direct calibration at the non-forested sites would almost certainly have improved simulations.

p6, line 18

I do not think that wind direction is needed to force the snow models, and please clarify whether any of them use surface temperature.

p7, line 1

"both liquid and solid fractions" means that total precipitation is measured, not separate snowfall and rainfall.

p7, line 23

After reading this, I expected Appendix A to give details of the vertical gradients used for temperature and precipitation interpolation.

p8, Table 1

Information on the elevation of the reanalysis grid points would be interesting here or in the text. Also, how much gap-filling was required in the station data?

p9

I am confused by SWIN-CLS. If R is measured radiation and SWIN is modelled clear-sky radiation, I don't see where the MSG cloud masks are being used. If R is incident solar radiation in cloudy conditions, isn't equation 2 the wrong way round?

p9, 13

Linear interpolation of sampled radiation fluxes rather than solar elevation-based inter-polation of accumulated fluxes will be biased. How do average fluxes compare? (briefly mentioned in 5.3 and turns out to be a source of error)

p10

Is the partitioning of total precipitation into snowfall and rainfall only applied to station measurements (in which case I expected to read about it in section 3) or also to the reanalyses, even though they provide separate snowfall and rainfall?

p12, line 31

Even a model that could account for impurities would not do so in this case because dust deposition was not provided as an input.

p18, line 33

MeteoIO errors are relatively small for temperature and snowfall, but errors in other forcing variables are not shown.

Correct spelling of "systematically" throughout

---

## Author Comment (AC2) · 21 Feb 2020

**Reply to Referee #1**

We thank the Reviewer for his thoughtful review and comments.
The reviewer's comments are reported below in bold font while our replies are in regular text:

**General comments: The authors of the manuscript (ms.) have tested six non-calibrated snow models at one mountain location by varying the time-resolution and origin of model forcing. The quality of meteorological forcing is indeed an important element in snow modelling, and the authors have here examined the sensitivity of snow model performance to varying input data quality. The manuscript is quite well-written, and the illustrations are mostly clear and understandable. The research idea is based on straightforward model testing, i.e. running an ensemble of models with different forcing data and calculating statistics on how the models' performance (evaluated against point snow observations) vary. However, the two main weaknesses of the ms. are in my opinion that:**
1) **the model evaluation is made at only one single site (Torgnon)**
2) **there is no discussion or treatment of uncertainty in solid precipitation measurements, which directly affect snow amounts in the comparison.**

**Although the ms. may provide an interesting case study for those interested in the specific models and the local site, the ms. does not have, in my opinion, large enough impact and interest for a wider audience to warrant publication in HESS. After reading the ms. twice, I felt in the end that I did not get much relevant information out of it for my modelling work, in another place, another country. As the authors themselves state (p. 22, lines 8-9): "This study offers some hints on this research topic".**

We reply to the two points highlighted by the reviewer in the text below, as these comments have been reported and expanded by the reviewer in the "Specific comments" section (see below). In particular, regarding the issue of the interest of the paper for a wider audience not specifically working on the models and site considered in this study, we address this in detail in our replies (mainly 1. and 5.) to better clarify the aims of our work, the results that can be exported to other snow models and sites, and the still-open issues, left for future investigation.

**Specific comments:**

1. **As snow is often spatially very inhomogeneously distributed, normally the utility in snow modelling is to get a grasp of this spatial variability. Thus, simulating snow in just one point has limited relevance, mostly restricted to snow process studies. In another point, the authors' results and model ranks might be changed significantly. As the authors themselves state, on p. 2, line 15: "Snow models are generally evaluated at a number of sites"; on p. 26, lines 2-3: "Further analysis at other test sites would be useful to explore the extent to which our results could be generalized to different situations or models".**

We see the reviewer's point and we are aware that the choice of a single study-site can be highlighted as a limitation, however we have some motivations to support this choice.
The strengths of our work are, in our opinion, the analysis of a multi-model ensemble, representative of different degrees of complexity of snow models, and the analysis of a wide range of possible meteorological forcing datasets, to explore in detail the response of the models to forcings with

different characteristics and resolutions in time and space. When planning this large, collaborative experiment, we carefully considered the choice of the site where to perform our analysis. The site we finally selected is quite unique as it provides high quality data in particular for precipitation (in most cases poorly measured in high elevation sites) and it is affected by low wind speeds, so that the snow-drift effect is limited. The combination of these two conditions is rare in high-elevation mountain measurement sites, nevertheless it is essential if we want to reduce the uncertainties on the input data. Repeating this effort in multiple test sites, for example in other alpine sites at different elevations and latitudes, or at non-alpine sites, i.e. in the Arctic) would certainly expand the vision provided by the paper but at the cost of larger uncertainties in the forcings which propagate across the modeling exercise and complicate the interpretation of the model outputs. Reducing the uncertainty on the "control" forcings is a prerequisite, in our case, to better separate the error due to model structure from the error due to the forcing. In this context, the selected site has represented for us the most appropriate benchmark for the aim of the paper. Extending the investigation to other test sites with less "optimal" forcing would be of great interest but, in our opinion, it should be addressed in a separate paper.

We hope to have clarified the motivations underlying our choice. We believe that this study, shedding light on the impacts of the model complexity and of the accuracy of the forcing on snow simulations, could be of interest for the readers of Hydrology and Earth System Sciences involved in catchment hydrology, snow modelling and snow and water resources management.

**2. The authors note on p.4 line 1-12: "the uncertainty on snow simulations due to the forcing can be comparable to or even larger than the uncertainty". They also refer on p.7 line 2 to Kochendorfer et al. (2017), who assess and provide algorithms to deal with the undercatch of solid precipitation. However, no effort is made to discuss, assess or correct the precipitation measurements at Torgnon station for the undercatch and/or examine the sensitivity of the authors' results for the inherent uncertainties in the observation-based model precipitation input (their CTL experiment with "optimal forcing").**

Following the reviewer's suggestion we analyzed in more detail the uncertainty associated with the observed precipitation and in particular the undercatch of snow which is common in mountain areas. The primary cause for snow precipitation undercatch is related to wind speed, with the amount of precipitation measured by a precipitation gauge relative to the actual amount of precipitation decreasing with increasing wind speed.

We quantified the wind-induced precipitation measurements errors by applying the method described in Kochendorfer et al. (2017). This method, derived by comparing precipitation measurements from unshielded and shielded (reference) gauges, consists in calculating a catch efficiency (CE), function of *air temperature* and *wind speed,* so that its inverse ($CE^{-1}$) can be used to correct actual precipitation data. The method has been specifically developed for OTT Pluvio2 gauges, i.e. the same type as that used at the Torgnon site.

Figure 1 shows the cumulated total precipitation at the Torgnon site measured by the precipitation gauge (black) compared to the precipitation adjusted with the Kochendorfer method (blue).

[Figure]

*Figure 1 Cumulated total precipitation at the Torgnon site measured by the OTT Pluvio2 precipitation gauge (black) compared to the precipitation adjusted with the Kochendorfer method (blue).*

The adjusted cumulated total precipitation exceeds the measured precipitation by 16% in average over the 5 snow seasons.

As the correction of total precipitation directly affects the amount of solid precipitation, we tested the effects of such correction on snow model simulations. We performed an additional experiment (CTL_prc-adj) in which the model forcing is the same as in the CTL run except for total precipitation, which is now *adjusted*, and snowfall which is now calculated from the *adjusted* total precipitation.

Figure 2 shows the results for the SNOWPACK model, and it displays the simulated snow depth (upper panel) and snow water equivalent (bottom panel) obtained in the CTL and in the CTL_prc-adj runs compared to observations.

In all snow seasons the snow depth and the snow water equivalent are remarkably overestimated in the CTL_prc-adj experiment compared to both observations and the CTL run. The additional snowfall input derived from the precipitation adjustment leads to an excess of snow accumulation on the ground which can be quantified in an average snow depth bias of 0.17 m compared to the 0.001 m bias in the CTL run. The RMSE is double in the CTL_prc-adj run compared to the CTL run (Table 1).

Given that the precipitation adjustment method itself is affected by its own uncertainties, and given that the application of the precipitation adjustment leads to a worsening in the snow model performances, we prefer to employ the original precipitation measurements as forcing in the snow model experiments. The discussion of the uncertainty of precipitation measurements and the effect of the precipitation adjustment on snow simulations has been included in the Appendix of the revised manuscript and summarized in the main text.

[Figure]

*Figure 2 Snow depth (upper panel) and snow water equivalent (lower panel) simulated by the SNOWPACK model when the adjusted total precipitation forcing is employed (CTL_prc-adj) compared to the control run (CTL) and observations.*

*Table 1. SNOWPACK model RMSE and bias for the simulated snow depth and snow water equivalent variables in the control run (CTL) and in the CTL_prc-adj experiment.*

|  | Snow depth | | SWE | |
| --- | --- | --- | --- | --- |
|  | RMSE [m] | BIAS [m] | RMSE [m] | BIAS [ m] |
| CTL | 0.10 | -0.001 | 0.04 | 0.02 |
| CTL_prc-adj | 0.20 | 0.170 | 0.10 | 0.09 |

**3. The authors claim that a bias adjustment of forcing data leads to more precise results (p.9. lines 29-31). This seems to me like a rather trivial point.**

The sentence in the manuscript reads "The last two experiments […] investigate if it is possible to improve the performances of snow models […] by applying two simple bias-correction methods to adjust air temperature and hence the amount of solid precipitation with respect to the total one."
The idea here is to check if:
- Correcting temperatures only (and keeping all the other variables unchanged except for solid and liquid precipitation whose partition depends on temperature) can improve the model snow simulations
- Very simple bias correction methods (such as the lapse rate correction and the subtraction of the mean bias) can be sufficient to improve model performances or more sophisticated techniques are necessary.

We rephrased the sentence in the text to clarify the meaning.

**4. The linear interpolation of shortwave radiation e.g. in the TIME-12h case, causing the large deviations of +97 W/m2 (p. 18, lines 3-4), is an unrealistically simplistic way to make the interpolation. In real modelling practice, I suppose most of us would use a sinus-curve form or something like that. Consequently, the issue here is more of poor modeling practice than lower time-resolution. This is only mentioned in section**

**Discussions (p.24, line 10), but would have been best to put into practice already in the authors' study.**

Following your suggestion we tested a more realistic way of estimating the 30 minute incoming shortwave radiation when only the measurements at 00:00 and 12:00 are available, i.e. as in the TIME-12h experiment. We employed the potential (clear-sky) incoming shortwave radiation (Knauer et al., 2018) at 30 minute temporal resolution and at the coordinates corresponding to the Torgnon station, and the surface station SWIN measurements at 12:00.

For each day of the year, the 48 daily values of potential radiation are rescaled according to the observed SWIN value at 12:00, to obtain an "estimated SWIN" (see Figure 3).

[Figure]

*Figure 3. Measured shortwave incoming radiation (SWIN) at the Torgnon site for the day 26 June 2017 (points), potential SWIN for that day (solid black line), "estimated SWIN" from the scaling of the potential SWIN on the value registered at h 12:00.*

The advantage of this method compared to the linear interpolation method is that the difference between the estimated and the observed SWIN radiation averaged over the full period is almost cancelled out , from +97 W/m2 when using the linear interpolation method to -0.87 W/m2 when using the method based on the scaling of the potential radiation.

In light of this result we run a new experiment TIME-12h-SWIN-POT, in which the forcing is the same as the one employed in the TIME-12h experiment except for the shortwave incoming radiation, which is now obtained with the potential radiation method.

Figure 4 shows the results of the TIME-12h-SWIN-POT experiment compared to that of the TIME-12h experiment, the CTL run and observations, for the SNOWPACK model and for the snow depth variable. The use of the potential radiation remarkably improves the agreement with observations, reducing the RMSE with respect to observations to a value which is comparable to the CTL run (Table 2).

The results of the TIME-12h-SWIN-POT experiment have been reported in the revised version of the manuscript and the effects of the two different interpolation methods (one based on the linear interpolation of the measurements and the other based on the scaling of the potential radiation) on the snow simulations have been discussed in the main text.

[Figure]

*Figure 4 Snow depth simulations obtained with the SNOWPACK model for the experiment TIME-12h-SWIN-POT compared to TIME-12h, the CTL run and observations.*

*Table 1 SNOWPACK model RMSE, BIAS and Pearson Correlation for the simulated snow depth in the CTL run, the TIME-12h and the TIME-12h-SWIN-POT experiments, compared to observations.*

|  | Snow depth | | |
|---|---|---|---|
|  | RMSE [m] | BIAS [m] | Pearson correlation |
| CTL | 0.10 | -0.001 | 0.97 |
| TIME-12h | 0.21 | -0.016 | 0.93 |
| TIME-12h-SWIN-POT | 0.11 | -0.036 | 0.97 |

**5. The point-specific biases and errors of the MeteoIO, GLDAS, ERA5 and ERA Interim, described in Sections 5.4 and 5.5 for the single Torgnon site, emphasize the weakness of this case study: the model evaluation results are difficult to generalize outside this site, where things and biases could be very different. Also, compensating errors may occur in the models which improve model performance. In other words, one gets "right results for wrong reasons" (p.13, lines 14-16; p. 21, lines 4-5; also p.24, line 17).**

We disagree that the biases and errors highlighted for MeteoIO, GLDAS, ERA5, and ERA-Interim at the Torgnon site emphasize the weaknesses of the work for two reasons. Concerning the presence

of biases, the aim of the paper is indeed to test how snow models respond to inputs which might be affected by large uncertainties and errors. Concerning the reviewer's remark on the difficulty of generalizing the results outside the area of study we addressed this point by testing the reanalysis products against observations over the Greater Alpine Region (GAR), to observe the spatial distribution of the temperature and precipitation biases and see if they are consistent at the mountain range scale.

ERA5, ERA-Interim and GLDAS temperatures have been averaged over the months October-June and over the years 1980-2014 (except for GLDAS which is available since 2000 only, so the averages have been calculated over the period 2000-2014), and then compared to the observational dataset EOBS (version 13, Haylock et al., 2008). EOBS is a daily gridded data set at 0.25° resolution, based on the European Climate Assessment and Data set station measurements.

[Figure]

*Figure 5 BIAS of ERA5, ERA-Interim and GLDAS air temperatures with respect to EOBS observations over the Greater Alpine Region. Temperatures have been averaged over the months from October to June and over the period 1980-2014 in the case of ERA5 and ERA-Interim, over the period 2000-2014 in the case of GLDAS. .*

ERA5 and GLDAS temperature biases are large and negative over the entire GAR (Figure 5). GLDAS bias is especially strong and it exceeds -4°C in most of the region. ERA5 bias is larger at high elevation than in lowlands. Compared to ERA5 and GLDAS, ERA-Interim temperature is in better agreement with observations, with mainly negative bias across the region and values close to zero (both positive and negative values) except at the mountain ridges in Western Alps.

Regarding precipitation, it is well known that standard surface station gauges have problems in capturing snowfall and thus they underestimate total precipitation in mountain areas. Similarly, also observational-based dataset such as EOBS have been found to suffer the underestimation of precipitation at high elevations (Turco et al., 2013). To overcome this problem, instead of using observation-based datasets as a reference, we evaluate precipitation differences with respect to a reanalysis, which inherently takes into account orographic effects. Figure 6 shows the ERA5 and GLDAS October-to-June accumulated precipitation differences relative to ERA-Interim (ratio) over the periods 1980-2014 and 2000-2014 respectively (GLDAS is available since 2000). Also in this case ERA5 spatial pattern is homogeneous over the Alpine range, with ERA5 showing consistently more precipitation than ERA-Interim in the mountain areas. Concerning GLDAS, we need to clarify that, while working on this response to reviewers, we noticed an error in the method which we used to perform the temporal interpolation of the original data and to derive the 30-minute resolution precipitation forcing for the "GLDAS" experiment. The error has now been fixed and the snow model runs driven by the GLDAS reanalysis are now being repeated with the correct forcing. The updated results will be reported in the revised version of the manuscript. With this correction, moreover, we found a much better agreement of the GLDAS precipitation with both Torgnon station measurements and with the ERA-Interim reanalysis over the Greater Alpine Region (Figure 6, right panel).

Overall, this analysis providing information on the spatial variability of the temperature and precipitation biases in the reanalysis products over the Alpine region broadens the perspective beyond the specific case of the Torgnon site. The biases at the Torgnon site result generally coherent with those found at the mountain range scale, although the magnitude of the bias can vary across the region

and with the elevation. This analysis addresses the question of how the bias in the main forcing variables (temperature and precipitation) at the Torgnon site can be generalized at larger scale, and in particular over the Alpine region.

The main results of this analysis are reported in the "Discussion" section of the revised manuscript, while the plots are reported in the Appendix.

[Figure]

*Figure 6 ERA5 and GLDAS relative differences with respect to ERA-Interim for the October-June accumulated precipitation over the periods 1980-2014 and 2000-2014 respectively.*

**6. Normally, the authors' results would be compared to previous studies in the section "Discussion". However, this three-page section only has one (!) single reference to other published studies. It also repeats many things already pointed out previously in the ms.**

We thank the Reviewer for this comment, the Discussion has been extensively modified avoiding repetitions and including the comparison of our results with those obtained in similar studies previously published in literature.

**Other points:**

**7. p.7 lines 22-23: provide values for the vertical gradients used here.**

The information has been added in the text, thank you

**8. Fig. 2. The meaning of dashed line circles in the Taylor graph should be explained.**

Guidance on how to interpret Taylor diagrams has been added in the text, thank you.

**9. The Appendix is short (a little figure and table, and five lines of text) and could be easily added to main text in Section 3.2.**

In the revised version of the manuscript the Appendix has been expanded. It now includes i) the discussion on the uncertainty of the total precipitation measurements at the Torgnon station; ii) information on how the Meteo-IO interpolated data have been derived; iii) discussion of the biases of the reanalyses over the Greater Alpine Region.

**10. The ms. contains a lot of numbers in tables 3-4, and displaying them more readerfriendly, like in the nice Fig. 6, would be good.**

Thanks for the suggestion, however in this case we preferred to keep the numerical values of data in the classic table form.

**References**

Haylock, M., Hofstra, N., Klein Tank, A., Klok, E., Jones, P., and New, M.: A European daily high-resolution gridded data set of surface temperature and precipitation for 1950–2006, J. Geophys. Res.-Atmos., 113, D20119, https://doi.org/10.1029/2008JD010201, 2008

Knauer, Jürgen, Tarek S. El-Madany, Sönke Zaehle, and Mirco Migliavacca. "Bigleaf—An R package for the calculation of physical and physiological ecosystem properties from eddy covariance data." PloS one 13, no. 8 (2018).

Kochendorfer, J., Nitu, R., Wolff, M., Mekis, E., Rasmussen, R., Baker, B., Earle, M. E., Reverdin, A., Wong, K., Smith, C. D., Yang, D., Roulet, Y.-A., Buisan, S., Laine, T., Lee, G., Aceituno, J. L. C., Alastrué, J., Isaksen, K., Meyers, T., Brækkan, R., Landolt, S., Jachcik, A., and Poikonen, A.: Analysis of single-Alter-shielded and unshielded measurements of mixed and solid precipitation from WMO-SPICE, Hydrol. Earth Syst. Sci., 21, 3525–3542, https://doi.org/10.5194/hess-21-3525-2017, 2017.

Turco, M., Zollo, A. L., Ronchi, C., De Luigi, C., & Mercogliano, P. (2013). Assessing gridded observations for daily precipitation extremes in the Alps with a focus on northwest Italy. Natural Hazards & Earth System Sciences, 13(6).

---

## Author Comment (AC3) · 21 Feb 2020

**Reply to Referee #2, Richard L.H. Essery**

We thank Prof. Richard L.H. Essery for his thoughtful review and comments.
The reviewer's comments are reported below in bold font while our replies are in regular text:

**Although limited to a single site and short on mechanistic explanations, the evaluations of several models in simulating snow mass, depth and density with several forcing datasets in this paper are of value (another with similar aims that should be cited is https://journals.ametsoc.org/doi/full/10.1175/JHM-D-15-0013.1).**
We thank the reviewer for this suggestion, the citation has been included in the revised paper.

**Measurements of outgoing shortwave and longwave radiation are mentioned but not used in the model evaluations; these might provide more insight.**

We agree that the evaluation of modelled outgoing shortwave and longwave radiation could provide interesting additional insights, as suggested by the reviewer. These variables are not provided by all the models considered in this study: they are missing completely in the simplest model considered, S3M, while the SMASH model provides the outgoing longwave radiation only. Therefore, we evaluated the simulated outgoing shortwave and longwave radiation at the surface for all remaining models (SNOWPACK, GEOTOP, HTESSEL and UTOPIA).

Figure 7 shows the difference (left) and the scatterplot (right) of the simulated and observed daily-averaged outgoing shortwave radiation for the SNOWPACK, GEOTOP, HTESSEL and UTOPIA models in the CTL run.

All the models tend to moderately underestimate the outgoing shortwave radiation, with SNOWPACK and HTESSEL showing the best agreement with observations both in terms of bias and of coefficient of determination $R^2$, compared to GEOTOP and UTOPIA. Differences between the simulated and observed outgoing shortwave radiation are mainly dependent on the representation of the albedo. These results suggest to check, in the UTOPIA model, the albedo, which is function of surface temperature and snow age as well as in the HTESSEL model, which nevertheless provides better outgoing shortwave radiation estimates.

[Figure]

*Figure 1 Difference (left figure) and scatterplot (right figure) of the simulated and observed outgoing shortwave radiation (CTL experiment) at the Torgnon site. Only four out of the six models considered in the paper provide the outgoing shortwave radiation among the output variables.*

Similarly to Figure 7, Figure 8 shows the difference (left) and the scatterplot (right) of the simulated and observed daily-averaged outgoing longwave radiation for the SNOWPACK, GEOTOP, HTESSEL, UTOPIA and SMASH models in the CTL run. The simulation of the net longwave radiation mainly affects the representation of the snow-melt dynamics. Both SNOWPACK and SMASH underestimate the outgoing longwave radiation, causing an excess in the snowpack energy available for the melting. However, none of the two models remarkably underestimates the snow depth, so other mechanisms might compensate for this behavior. GEOTOP, HTESSEL and UTOPIA outgoing longwave radiation do not show systematic biases.

The considerations about the evaluation of the simulated outgoing shortwave and longwave radiation have been reported in the main text of the manuscript. We thank the reviewer for the suggestion.

[Figure]

*Figure 2 Difference (left figure) and scatterplot (right figure) of the modelled and observed outgoing longwave radiation (CTL experiment) at the Torgnon site. Only five out of the six models considered in the paper provide the outgoing longwave radiation among the output variables.*

- **p2, line 28 There is no need for feedbacks for early differences in snow simulations to persist throughout the winter if the imposed conditions remain too cold for melt.**

We thank the reviewer for the suggestion; the text has been modified accordingly.

- **p3, line 22 delete "air" in "open air sites"**

Done, thank you.

- **p3, line 25 Specifically, Rutter et al. (2009) found that benefits from calibration at forest sites did not transfer to nearby non-forested sites. Direct calibration at the non-forested sites would almost certainly have improved simulations.**

We thank the reviewer for the suggestion; the text has been modified accordingly.

- **p6, line 18 I do not think that wind direction is needed to force the snow models, and please clarify whether any of them use surface temperature.**

Indeed none of the models employs the wind direction, thanks for the correction. With "surface temperature" we actually meant "ground temperature at 2 cm depth", and this variable is needed by the SNOWPACK model only. We have now clarified both points in the text.

- **p7, line 1 "both liquid and solid fractions" means that total precipitation is measured, not separate snowfall and rainfall.**

Yes, exactly. At the Torgnon station the total precipitation amount is measured. We modified the text to better clarify it, thank you.

- **p7, line 23 After reading this, I expected Appendix A to give details of the vertical gradients used for temperature and precipitation interpolation.**

Yes, we added this information in the text, thanks for the suggestion.

- **p8, Table 1 Information on the elevation of the reanalysis grid points would be interesting here or in the text. Also, how much gap-filling was required in the station data?**

Information on the elevation of the reanalyses at the Torgnon gridpoint, as well as the % of missing value for each input variable provided by the Torgnon station have been added in Table 2.

- **p9 I am confused by SWIN-CLS. If R is measured radiation and SWIN is modelled clearsky radiation, I don't see where the MSG cloud masks are being used. If R is incident solar radiation in cloudy conditions, isn't equation 2 the wrong way round?**

Yes, the equation was wrong and it has been corrected, many thanks for pointing it out. MSG cloud mask is used to identify the radiometers under clouds and compute an average attenuation factor. We have better explained this in the text.

- **p9, 13 Linear interpolation of sampled radiation fluxes rather than solar elevation-based interpolation of accumulated fluxes will be biased. How do average fluxes compare? (briefly mentioned in 5.3 and turns out to be a source of error)**

We agree that linear interpolation of sampled shortwave radiation fluxes introduces errors in the forcing data, which can be large when the sampling time step is 12h (TIME-12h experiment). For this case, we tested a different method to estimate 30 minutes shortwave incoming radiation (SWIN) from 12h samplings, based on the rescaling of the potential radiation with respect to the measurement at 12:00.

The details of this exercise, as well as the comparison between the linearly interpolated SWIN and the modified SWIN forcing, are provided in the reply to Referee#1, at point 4. In summary, while the linearly interpolated SWIN forcing shows an average bias of +97 W/m$^2$ compared to observations in the period of investigation, the modified SWIN has significantly reduced the bias to a value close to zero (-0.87 W/m$^2$), so the average flux is conserved. We ran an additional experiment (TIME-12h-SWIN-POT) using the modified SWIN forcing (see details in the reply to Referee#1, point 4).

A detailed discussion of this point and the results of the new experiment have been included in the manuscript in Sections 4 and 5.3

- **p10 Is the partitioning of total precipitation into snowfall and rainfall only applied to station measurements (in which case I expected to read about it in section 3) or also to the reanalyses, even though they provide separate snowfall and rainfall?**

We applied the same method to separate rainfall and snowfall for all the forcing datasets, including reanalyses. We better clarified it in Section 3 and 4.

- **p12, line 31 Even a model that could account for impurities would not do so in this case because dust deposition was not provided as an input.**

Yes, thank you for this comment. We modified the text as suggested

- **p18, line 33 MeteoIO errors are relatively small for temperature and snowfall, but errors in other forcing variables are not shown. Correct spelling of "systematically" throughout**

The MeteoIO forcing biases with respect to the Torgnon measurements are relatively small on average not only for temperature and snowfall but for all variables (Figure 9). In order to explain the discrepancies between the simulated SWE and snow depth in the MeteoIO experiment and observations (see i.e. Table 4 in the manuscript), we investigated the temporal variability of the air temperature bias (Figure 10) and we related it to the simulated SWE and snow depth in the MeteoIO experiment (Figure 11). The temperature bias is about -1°C on average over the considered time period, however in winter the cold bias is generally stronger and it can reach values exceeding -4°C (Figure 10). Concerning MeteoIO-driven snow model simulations, the main issue is the overestimation of snow depth in winter (in selected snow seasons) and in spring (in all seasons). A plausible explanation for these errors is that colder-than-observed winter temperatures might favor the development of a cold snowpack which melts too slowly. Consequently, the models tend to overestimate the snow at surface and to predict a delayed ablation date.

We added these comments in Section 5.4 of the manuscript.

The typo "systematically" has been corrected, thank you.

*Figure 3. Scatterplot of the meteorological forcing of the MeteoIO experiment with respect to the CTL run.*

**MeteoIO Air temperature bias**

[Figure]

*Figure 4 MeteoIO air temperature bias with respect to Torgnon station measurements.*

**MeteoIO**

[Figure]

[Figure]

*Figure 5 Simulated snow depth (top) and snow water equivalent for the MeteoIO experiment compared to observations.*

---

## Author Response (AR2)

Dear Editor,
   please find enclosed the revised version of the manuscript taking into account the suggestions and remarks of the Reviewer. All the corrections to the text have been made and we improved the figures as suggested. We only left unchanged the gray color of the background of Figure 6 since it is an aesthetic choice which does not affect the readability of the figure.
Information on "Data availability" has also been included.
Many thanks for following  all the review steps of this manuscript.
Kind regards
Silvia Terzago and coauthors